# A broadband thermal emission spectrum of the ultra-hot Jupiter WASP-18b

Louis-Philippe Coulombe[1,2 ✉], Björn Benneke[1,2], Ryan Challener[3], Anjali A. A. Piette[4], Lindsey S. Wiser[5], Megan Mansfield[6], Ryan J. MacDonald[3,7,8], Hayley Beltz[3], Adina D. Feinstein[9], Michael Radica[1,2], Arjun B. Savel[10,11], Leonardo A. Dos Santos[12], Jacob L. Bean[9], Vivien Parmentier[13], Ian Wong[14], Emily Rauscher[3], Thaddeus D. Komacek[10], Eliza M.-R. Kempton[10], Xianyu Tan[15,16,17], Mark Hammond[17], Neil T. Lewis[18], Michael R. Line[5], Elspeth K. H. Lee[19], Hinna Shivkumar[20], Ian J. M. Crossfield[21], Matthew C. Nixon[10], Benjamin V. Rackham[22,23], Hannah R. Wakeford[24], Luis Welbanks[5], Xi Zhang[25], Natalie M. Batalha[26], Zachory K. Berta-Thompson[27], Quentin Changeat[28,29], Jean-Michel Désert[20], Néstor Espinoza[12], Jayesh M. Goyal[30], Joseph Harrington[31,32], Heather A. Knutson[33], Laura Kreidberg[34], Mercedes López-Morales[35], Avi Shporer[23], David K. Sing[36,37], Kevin B. Stevenson[38], Keshav Aggarwal[39], Eva-Maria Ahrer[40,41], Munazza K. Alam[4], Taylor J. Bell[42], Jasmina Blecic[43,44], Claudio Caceres[45,46,47], Aarynn L. Carter[26], Sarah L. Casewell[48], Nicolas Crouzet[49], Patricio E. Cubillos[50,51], Leen Decin[52], Jonathan J. Fortney[26], Neale P. Gibson[53], Kevin Heng[41,54,55], Thomas Henning[34], Nicolas Iro[56], Sarah Kendrew[28], Pierre-Olivier Lagage[57], Jérémy Leconte[58], Monika Lendl[59], Joshua D. Lothringer[60], Luigi Mancini[34,50,61], Thomas Mikal-Evans[34], Karan Molaverdikhani[34,54,62], Nikolay K. Nikolov[12], Kazumasa Ohno[26], Enric Palle[63], Caroline Piaulet[1,2], Seth Redfield[64], Pierre-Alexis Roy[1,2], Shang-Min Tsai[65], Olivia Venot[66] & Peter J. Wheatley[40,41]

Close-in giant exoplanets with temperatures greater than 2,000 K ('ultra-hot Jupiters') have been the subject of extensive efforts to determine their atmospheric properties using thermal emission measurements from the Hubble Space Telescope (HST) and Spitzer Space Telescope[1–3]. However, previous studies have yielded inconsistent results because the small sizes of the spectral features and the limited information content of the data resulted in high sensitivity to the varying assumptions made in the treatment of instrument systematics and the atmospheric retrieval analysis[3–12]. Here we present a dayside thermal emission spectrum of the ultra-hot Jupiter WASP-18b obtained with the NIRISS[13] instrument on the JWST. The data span 0.85 to 2.85 μm in wavelength at an average resolving power of 400 and exhibit minimal systematics. The spectrum shows three water emission features (at >6σ confidence) and evidence for optical opacity, possibly attributable to H−, TiO and VO (combined significance of 3.8σ). Models that fit the data require a thermal inversion, molecular dissociation as predicted by chemical equilibrium, a solar heavy-element abundance ('metallicity', M/H = $1.03^{+1.11}_{-0.51}$ times solar) and a carbon-to-oxygen (C/O) ratio less than unity. The data also yield a dayside brightness temperature map, which shows a peak in temperature near the substellar point that decreases steeply and symmetrically with longitude towards the terminators.

The thermal emission spectra of ultra-hot Jupiters typically have muted spectral features and thus closely resemble blackbodies in existing narrowband measurements[2,3]. The interpretation of these spectra has been controversial, with some studies claiming that the data are indicative of high metallicities and C/O ratios[4,6]. Alternatively, other studies have proposed that approximately solar-composition models including the effects of molecular dissociation and continuum opacity from the H− ion can match the data[7–10,14]. The ultra-hot Jupiter WASP-18b has been a subject of this controversy. Past HST and Spitzer Space Telescope secondary-eclipse and phase-curve observations have found

high dayside temperatures[15,16], indicative of low heat redistribution potentially caused by magnetic drag[16], weak or no spectral features from H₂O in the Hubble bandpass and signs of a temperature inversion in the broadband Spitzer photometry[4,7,11]. High-resolution observations of the dayside of WASP-18b have detected CO, OH and H₂O at signal-to-noise ratios of 4.0, 4.8 and 3.3, respectively[17]. The molecular features were observed in emission, indicative of a thermal inversion, although the lack of a spectral continuum led to poor constraints on the temperature of the atmosphere. The metallicity and C/O ratio values retrieved from the high-resolution data are consistent with solar but

also depend on the physical assumptions, with the metallicity constraints ranging from 1 to 100 times solar between the self-consistent and free-chemistry analyses.

We observed a secondary eclipse of WASP-18b with NIRISS/SOSS[13] as part of the JWST Transiting Exoplanet Community Early Release Science Program[18]. WASP-18b is a 10.4 ± 0.4 $M_J$ ultra-hot Jupiter on a 0.94-day orbit around a bright ($J$ mag = 8.4) F6V-type star[19]. Our goals were to characterize the atmosphere of WASP-18b and demonstrate the capabilities of JWST observations for exoplanets orbiting bright stars. We used the SUBSTRIP96 subarray mode (96 × 2,048 pixels) to avoid saturation by minimizing the individual integration times. The SUBSTRIP96 mode covers the first spectral order between 0.85 and 2.85 μm. The time series spans 6.71 h and consists of 2,720 continuous integrations with three groups and 8.88 s per integration, delivering an integration efficiency of 67%. We used the F277W filter in the final ten integrations to check for contamination from background stars and found none. We observed for 2.83 h before the eclipse and continued for 1.70 h after the eclipse. The observations captured 107° of the orbit of WASP-18b. Assuming it is tidally locked, the planet rotated by the same angle during the observation.

We analysed the data using four independent pipelines: NAMELESS, nirHiss, supreme-SPOON and transitspectroscopy (see Methods). Beginning from the raw calibrated data or stage 1 products, we performed custom reductions and extracted 1D spectra from each integration using either a fixed-width aperture (NAMELESS, supreme-SPOON and transitspectroscopy) or an optimal extraction (nirHiss) technique. We put particular emphasis on the removal of 1/$f$ noise ($f$ is frequency), a signal with power spectrum inversely proportional to the frequency that is introduced through variations of the reference voltage as the detector is being read. Its removal requires careful treatment, as the spectral trace covers most of the SUBSTRIP96 subarray (see Methods). Finally, we obtained spectrophotometric light curves by summing the observed flux within 408 spectral bins, each containing five pixel columns on the detector (Extended Data Fig. 1). We also produced a white-light curve by summing the spectrophotometric light curves over all wavelengths. All light curves show a sudden decrease in flux around the 1,336th integration, simultaneous to a tilt event from one of the segments of the primary mirror[20,21]. In the NIRISS data, this can be independently identified through a small but detectable morphological change in the spectral trace on the detector (Extended Data Fig. 2). We also observed small variations in the spectral trace morphology throughout the time series, mainly of its position and full width at half maximum, that are correlated with the measured flux. We detrended against these morphological changes at the fitting stage.

We analysed the extracted white and spectrophotometric light curves by fitting the parameters for the secondary eclipse, the partial phase curve and the systematics using ExoTEP[22] (see Methods). When fitting the white-light curve, we allowed the semi-major axis and impact parameter to vary, constrained by Gaussian priors that were derived from an analysis of the full-orbit phase curve observed by the Transiting Exoplanet Survey Satellite (TESS; see Methods). We imposed a uniform prior on the mid-eclipse time and assumed a circular orbit. Those parameters were subsequently fixed when fitting the spectrophotometric bins (see Methods). The maximum star plus planet signal-to-noise ratio for a single-pixel spectrophotometric light curve is 617 at 1.14 μm, with the signal-to-noise ratio curve closely following the shape of the throughput-weighted stellar spectrum and reaching a minimum of 62 at 2.83 μm. All four reductions yield consistent results (Extended Data Fig. 3), with all resulting thermal emission spectra being consistent at less than one standard deviation on average. The residuals for each spectrophotometric light curve closely follow the expected 1/√$n$ ($n$ is the number of events) scaling of Poisson noise when binned in time and the white-light curve bins down to 5 ppm over 1-h timescales (see Extended Data Fig. 4).

The secondary-eclipse spectrum was created by collating the planet-to-star flux ratio values at mid-eclipse for all the wavelength channels (see Fig. 1). We then multiplied by a PHOENIX stellar spectrum model[23] produced using previously published parameters for WASP-18 (that is, $T_{eff}$ = 6,435 K, log $g$ = 4.35 and [Fe/H] = 0.1 (ref. 24)) to convert the dayside secondary-eclipse spectrum into the thermal emission spectrum of the planet (see Fig. 1). For clarity, we also computed the brightness temperature spectrum, commonly used in planetary science, by calculating the blackbody temperature corresponding to the observed thermal emission in each wavelength bin (see Fig. 2). This transformation into brightness temperature facilitates identification of the various opacity sources by removing the large average slope caused by the behaviour of the Planck emission across the NIRISS/SOSS wavelength range.

The observed brightness temperature spectrum shows strong deviations from a blackbody. It is dominated by the 1.4-μm, 1.9-μm and 2.5-μm water emission features and a rise in brightness temperature shortwards of 1.3 μm. The rise in brightness is caused by the combined opacities of $H^-$, TiO and VO, and we infer a combined detection significance of 3.8$\sigma$ for these three species (Fig. 2). All molecular features appear in emission, indicating a thermal inversion (that is, temperature increases with altitude; see also Fig. 3). The water features are consistent with a solar-composition atmosphere, as predicted by 1D radiative–convective models and 3D general circulation models (GCMs). They are strongly inconsistent with any high-C/O or high-metallicity scenarios[4] (Fig. 2b), solving the tension from past HST observations[4,7] that obtained inconsistent results owing to differences in model assumptions and the limited bandpass of HST/WFC3. This finding is further strengthened by the lack of detectable CO features at 1.6 and 2.4 μm, which should be the dominant species in a high-metallicity, carbon-rich atmosphere (Fig. 2b). Using the free retrieval, we constrain the 3$\sigma$ upper limit of the CO log mixing ratio to −2.42 (see below; Extended Data Fig. 5).

Quantitatively, we infer an atmospheric metallicity of $0.82^{+0.59}_{-0.37}$ times solar when fitting the NAMELESS reduction to a grid of self-consistent 1D radiative–convective models and, consistently, $1.19^{+1.22}_{-0.67}$ times solar when allowing for a free vertical temperature structure in the chemically consistent retrievals. Both modelling approaches accounted for the thermal dissociation of water in the upper atmosphere and assumed chemical equilibrium. In both cases, the best fits are obtained for sub-solar C/O values around 0.03–0.30, for which the solar C/O value is 0.55 (ref. 25). The self-consistent models give a 3$\sigma$ upper limit of 0.2, whereas the free-temperature-structure retrieval allows C/O values up to 0.6 at 3$\sigma$ (Fig. 3), consistent with the upper limit from high-resolution dayside thermal emission observations[17]. We also assessed the effect of disequilibrium chemistry (see Methods) on the observed thermal emission and found the impact to be below 10 ppm, owing to the short chemical timescales in this hot atmosphere, justifying the assumption of thermochemical equilibrium models. Also, we performed a free-chemistry atmospheric retrieval[11,26], including the effects of thermal dissociation[9], and inferred a $H_2O$ deep atmospheric log mixing ratio of $-3.23^{+0.45}_{-0.29}$, consistent with the models assuming chemical equilibrium (Fig. 2c) and the solar value of −3.21 (ref. 27). We identify a strong thermal inversion with a temperature increase of 500 K in the middle atmosphere from 1 bar to 0.01 bar, which corresponds to the pressure range covered by the contribution functions (Fig. 3a). Our best-fit radiative–convective model provides strong evidence that the temperature inversion is caused by the absorption of stellar light by TiO (see Extended Data Fig. 6). At first sight, this can seem at odds with high-spectral-resolution observations that have detected other species able to create thermal inversions, such as atomic iron[10], but have had trouble detecting TiO (ref. 28). This tension is easily solved when considering that both TiO and water thermally dissociate in the upper atmospheric layers of ultra-hot Jupiters. Our observations, on the other hand, are sensitive to deeper layers of the atmosphere close to the infrared photosphere, which extends from 0.01 down to 1 bar (see contribution function on Fig. 3a), in the region in which molecules such as water and TiO

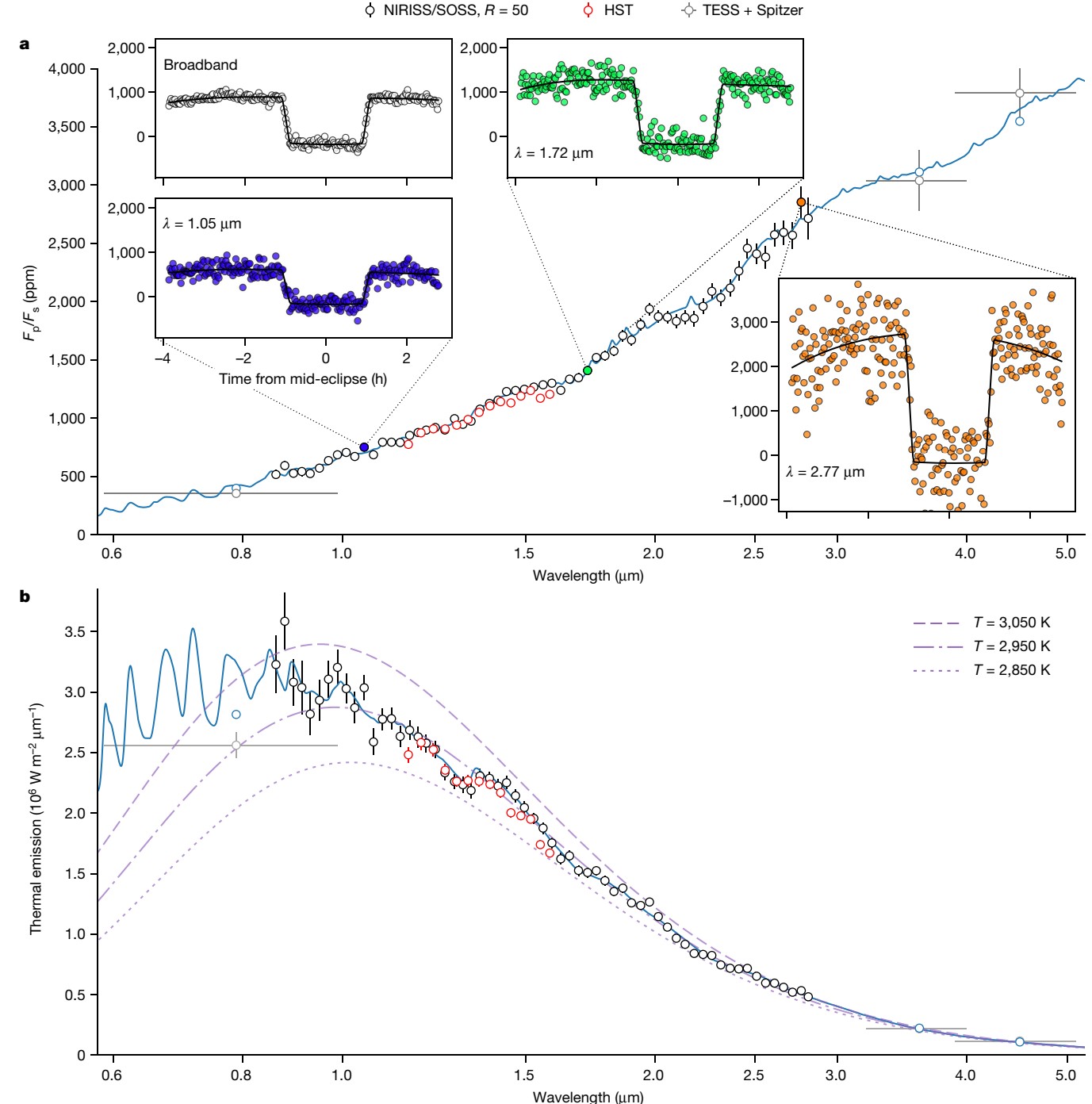

**Fig. 1 | Dayside thermal emission spectrum of WASP-18b. a**, Observed dayside planet-to-star flux ratio spectrum (black points) and their 1σ error bars, binned at a fixed resolving power of $R = 50$ for visual clarity. Past HST[7] (red points), TESS (see Methods) and Spitzer[15] (grey points) are shown for comparison. We show the best-fit model (blue line) from the SCARLET chemical-equilibrium retrieval, extrapolated to the TESS and Spitzer wavelengths considering the same atmospheric parameters. We find that the measured spectrum is in good agreement with the past HST observations. The throughput-integrated model is shown for the TESS and Spitzer points (blue points). The white (broadband) light curve (white points) and three example spectrophotometric light curves

(blue, green and orange points at 1.05, 1.72 and 2.77 μm, respectively), along with their best-fitting models (black line), are shown to scale. The phase variation of the measured planetary flux around the secondary eclipse is clearly visible. **b**, Planetary thermal emission spectrum of WASP-18b, as computed from the $F_p/F_s$ spectrum and the PHOENIX stellar spectrum. The shortest wavelengths of the NIRISS/SOSS first order reach the maximum of the planetary spectral energy distribution, thereby enclosing 65% of the total thermal energy emitted by the planet. Blackbody spectra for temperatures $T = 2,850$ K (dotted line), 2,950 K (dash-dotted line) and 3,050 K (dashed line) are shown in purple, with the best-fitting blackbody spectrum to the NIRISS data being $T = 2,950 \pm 3$ K.

recombine (see Extended Data Fig. 9). Even though our model also predicts that iron can produce a thermal inversion, its near-constant abundances mean that inversions owing to iron happen at pressures lower than 1 millibar and not where we detect the main thermal inversion.

The precise constraints on the atmospheric metallicity and C/O, measured by probing the deep atmosphere of WASP-18b, which is unaffected by thermal dissociation, enable us to investigate possible formation scenarios of WASP-18b. Considering the core-accretion formation

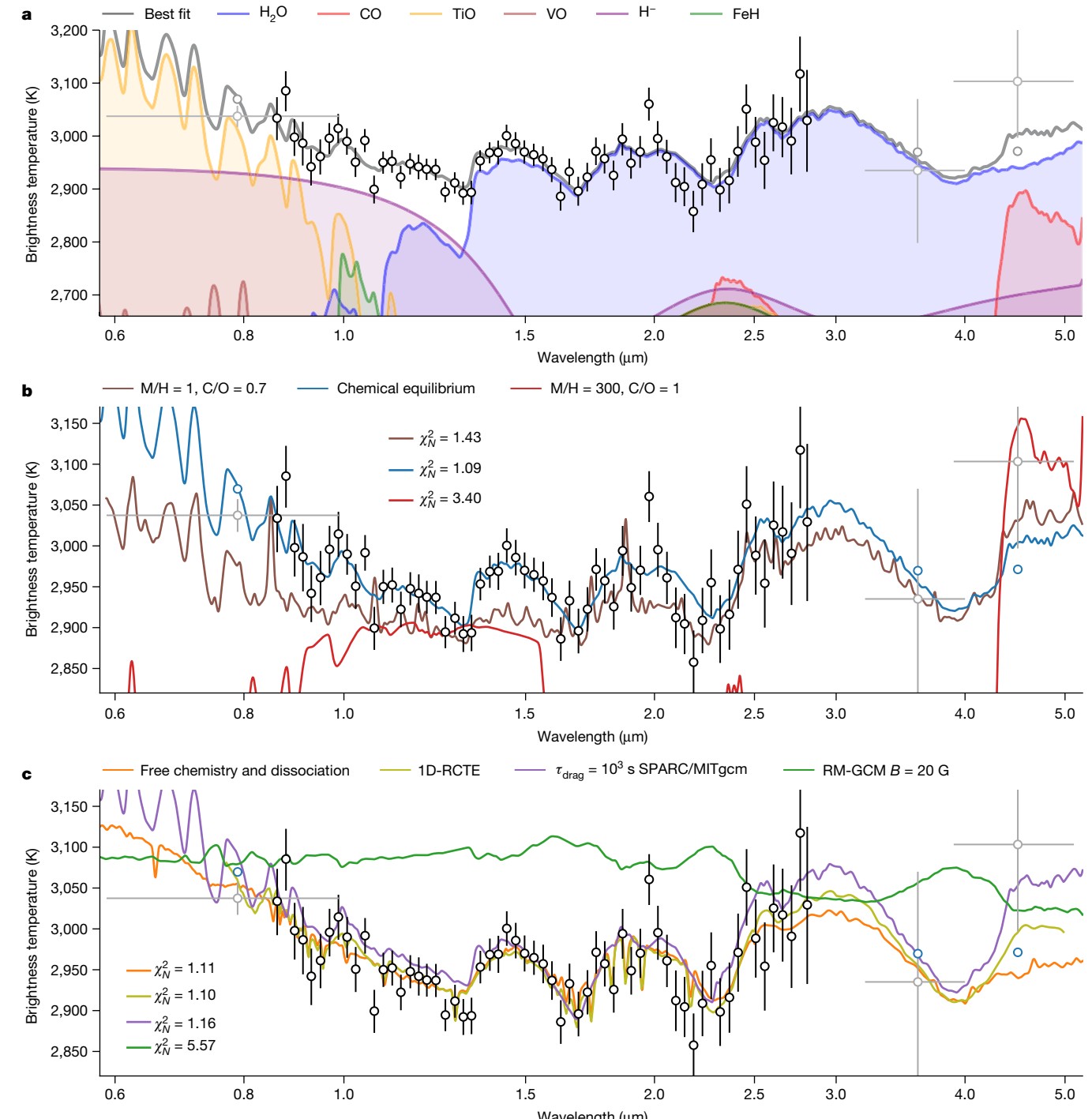

**Fig. 2 | Brightness temperature spectrum of WASP-18b. a**, Brightness temperature of WASP-18b as a function of wavelength, with models extrapolated to the TESS and Spitzer points considering the same atmospheric parameters. All data are plotted with their 1$\sigma$ error bars. The H$_2$O emission features at 1.4, 1.9 and 2.5 μm are clearly visible. The increase in brightness temperature observed in the water features is indicative of a thermal inversion. We also observe a downward slope in the spectrum from 0.8 to 1.3 μm as the opacities of H$^-$, TiO and VO decrease. We find that the precision of the observations at 2.4 μm is not sufficient to detect the small expected contribution from CO. **b**, Comparison of the high metallicity and C/O case[4] (red), as well as the solar metallicity case with

H$^-$ opacity and H$_2$O dissociation[7] (brown, best fit to the HST data shown in Fig. 1) that could both explain the past HST observations. We also show the SCARLET best-fit model to the NIRISS observations (blue). **c**, Median fits of the free-chemistry retrieval (orange) and of the self-consistent chemical-equilibrium grid retrieval (green). We also show the dayside spectra obtained by post-processing the SPARC/MITgcm (purple) and RM-GCM (green) for a drag timescale of $\tau_{drag} = 10^3$ s and a magnetic field strength of $B = 20$ G, respectively. We find that the SPARC/MITgcm better reproduces the observed features, as the RM-GCM is more isothermal.

scenario[29], the measured atmospheric metallicity of WASP-18b, consistent with the near-solar metallicity of the host star WASP-18 ([Fe/H] = 0.1 ± 0.1)[19,30], indicates that accretion of protoplanetary gas,

rather than rocky or icy planetesimals, dominated the late-stage formation of the planet. The mass–metallicity trend derived from solar system planets[27,31–33] predicts that the metallicity decreases as the mass

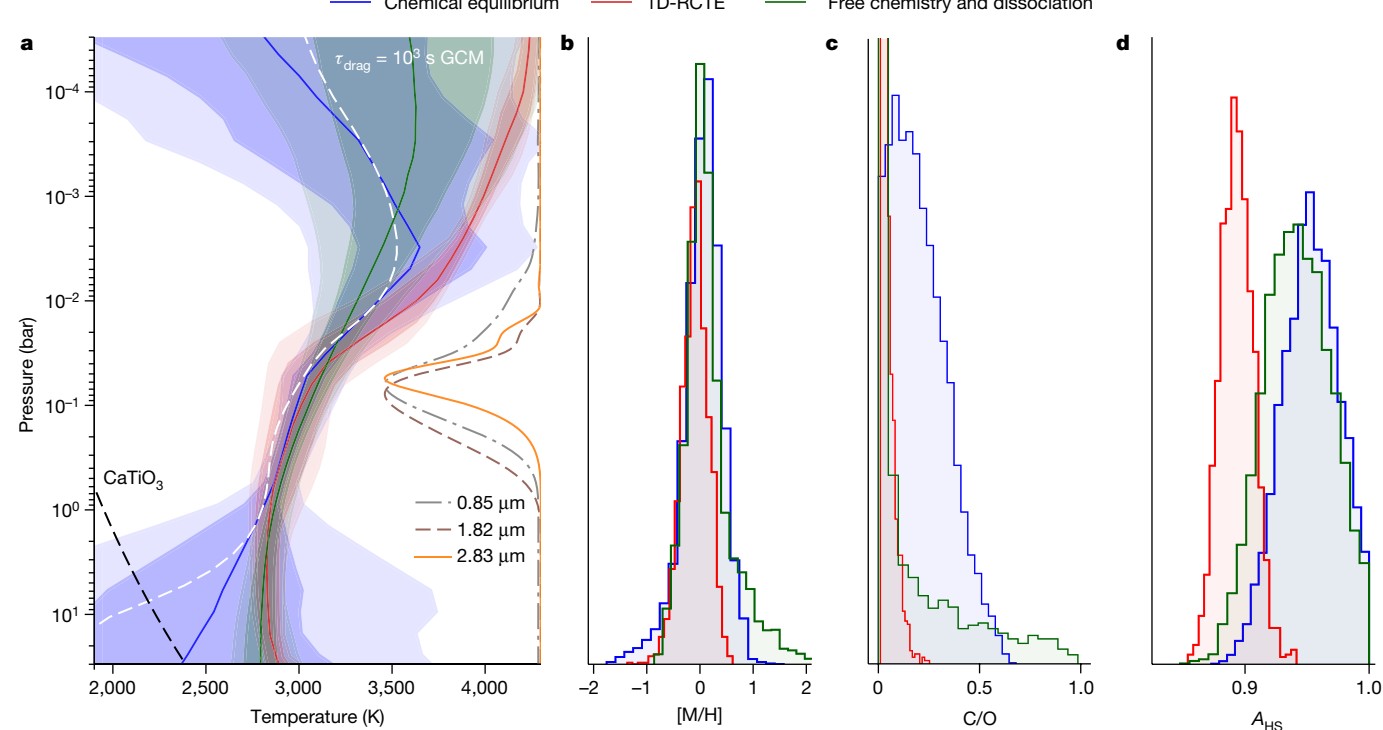

**Fig. 3 | Atmospheric constraints from the chemical-equilibrium and free-chemistry retrievals. a**, Retrieved temperature–pressure profiles with 1σ and 2σ contours for the chemical equilibrium with free temperature–pressure profile (blue), radiative–convective–thermochemical equilibrium (1D-RCTE, red) and free chemistry with thermal dissociation (green) retrievals. The retrieved temperature–pressure profiles are consistent between the retrievals and show an inversion in the pressure range that is constrained from the observations, as shown by the contribution functions at 0.85 (dot-dashed grey line), 1.82 (dashed brown line) and 2.83 μm (orange line). The temperature–pressure profile of WASP-18b is above the CaTiO3 condensation curve[50] (dashed black line) at almost all pressures, which motivates the presence of a temperature inversion caused by TiO, as Ti is available in gas form. The dayside average temperature–pressure profile of the $\tau_{drag} = 10^3$ s SPARC/MITgcm (dashed white line) is computed from the viewing angle average of $T(P)^4$ and shown for comparison. **b–d**, We also show the posterior probability distributions of the atmospheric metallicity [M/H] (**b**), C/O ratio (**c**) and area fraction $A_{HS}$ (**d**). The area fraction $A_{HS}$ is a scaling factor applied to the thermal emission spectrum to compensate for the possible presence of a concentrated hotspot contributing to most of the observed emission[51]. All methods retrieve metallicities consistent with solar at 1σ. The retrieved C/O 3σ upper limits are of 0.6 and 0.2 for the chemical equilibrium with free temperature–pressure profile and the 1D-RCTE retrievals, respectively. Finally, we find that the area fraction $A_{HS}$ is consistent with 1 when allowing the temperature–pressure profile to vary freely, indicating the lack of a concentrated hotspot on the dayside contributing to most of the observed emission.

of the planet increases, approaching the composition of the star for the most massive planets. Our finding of solar metallicity, three times lower than that of Jupiter, is consistent with this trend, given the mass of WASP-18b of 10.4 $M_J$. Assuming that it formed at solar metallicity, we find that up to 181 $M_⊕$ of metals could have been accreted during the migration of WASP-18b before it exceeds the 2σ upper limit on the metallicity obtained from the atmospheric retrieval (see Methods). This quantity of metals almost certainly exceeds the amount that is available in the disk for accretion during migration. The metallicity that is measured for WASP-18b is therefore probably representative of the bulk composition of the protoplanetary disk at the formation location of the planet. Furthermore, the low C/O ratio disfavours forming WASP-18b beyond the $CO_2$ ice line followed by an inward migration after the disk has dispersed, as gas accretion in that region would have led to high C/O values[34]. Detailed spectroscopic observations of the 4.5-μm CO feature, which is found within the spectral range of JWST NIRSpec G395H, could lead to a more stringent constraint on the C/O ratio and, thus, on the formation and migration history of WASP-18b. A detailed interpretation of the atmospheric C/O ratio of WASP-18b would require knowledge of the C/O ratio of the host star. This was recently found to be markedly subsolar (C/O = 0.23 ± 0.05)[30] based on high-resolution spectroscopy. However, stellar C/O measurements are especially challenging owing to stellar model inaccuracies and weak/blended absorption lines[35], so further confirmation is warranted.

Another possible formation scenario for WASP-18b is through collapse of the disk from gravitational instability[36] with a disk-free migration. This process leads to an atmospheric metallicity and C/O dictated by the local disk composition and is expected to result in planets with stellar-to-superstellar metallicities and substellar-to-stellar C/O (ref. 37), in agreement with our results.

As well as the extracted planetary spectrum and the elemental abundances, we also recover the broadband brightness temperature distribution across the dayside of WASP-18b using the eigencurves eclipse-mapping method (see Methods and refs. 38–40). We begin this analysis with the systematics-corrected white-light curve. We performed two independent applications of the method, both enforcing positive flux contribution from visible locations on the planet. We find two brightness map solutions that fit the data similarly well (Fig. 4). We convert brightness maps to brightness temperature maps assuming a PHOENIX spectrum[41] for the star at 6,432 ± 48 K, log $g$ = 4.35 ± 0.05 (ref. 24) and $R_p/R_s$ = 0.09783 ± 0.00028 (Extended Data Table 1). The first solution (blue model) shows a brightness temperature plateau stretching from approximately −40° to +40° of longitude relative to the substellar point, with a virtually constant latitudinally averaged brightness temperature of $3,124^{+35}_{-5}$ K. The second solution (red model) shows a more concentrated hotspot at the substellar point with a maximum brightness temperature of $3,272^{+9}_{-12}$ K and a consistent decrease in temperature both eastward and westward of the substellar

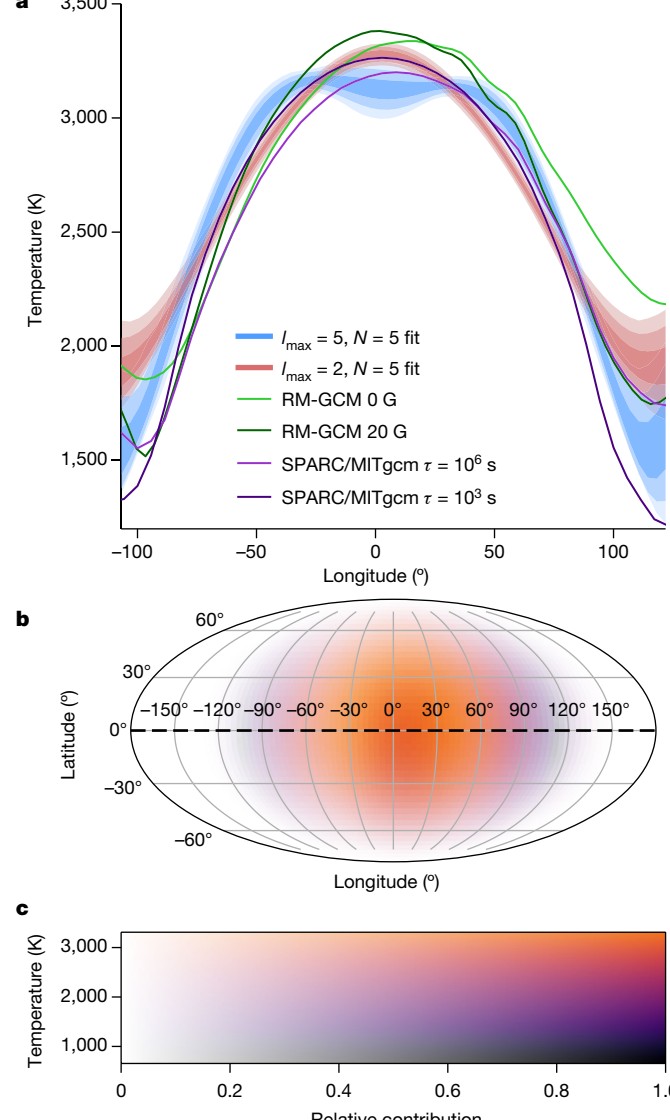

**Fig. 4 | Retrieved temperature map of WASP-18b. a**, Latitudinally averaged brightness temperatures (see Methods) of the planet along the equator. The blue and red shaded areas show solutions for $l_{max} = 5$, $N = 5$ and $l_{max} = 2$, $N = 5$ (see Methods), respectively. The effective broadband wavelength of the map, weighted by the observed flux ($F_p + F_s$) and instrument response, is $\lambda = 1.27\,\mu m$. Statistically, the blue model is marginally preferred. Dark, medium and light shading denote the $1\sigma$, $2\sigma$ and $3\sigma$ confidence regions, respectively, showing the range of model possibilities. Overplotted are several predictions from GCMs with magnetic field[52] (green) or uniform[53] (purple) drag timescales (see Methods). The plot only shows longitudes emitting at least 10% of the substellar flux. **b**, The temperature map of WASP-18b for the $l_{max} = 5$, $N = 5$ solution. Along the equator at $-90°$, $0°$ and $90°$ longitude, the temperatures are 1,744 K, 3,121 K and 2,009 K, respectively. **c**, The colour bar for the map shown in **b**. Colour represents the brightness temperature and saturation represents the relative contribution to the light curve based on its visibility.

point. Both solutions consistently reveal a steep temperature drop with longitude towards the terminators, with the inferred brightness temperature falling to $1{,}913^{+87}_{-27}$ K at the western terminator and $2{,}129^{+63}_{-11}$ K at the eastern terminator (blue model), and neither shows any substantial shift of the brightest region away from the substellar point. This is consistent with what was measured from the HST phase curve of WASP-18b (ref. 16). The high temperatures covering most of the dayside in both solutions, along with the steep decrease in temperature near the limbs, are consistent with the atmospheric-retrieval results (Fig. 3d).

Beyond the terminators and leading to the nightside, we infer a continued drop in the thermal emission. Our JWST observations have the sensitivity to examine part of the nightside because the planet rotates by 107° during the time series, providing a view of up to 62.5° of the nightside east of the eastern terminator at the beginning and ending 44.5° west away from the western terminator. The lack of a notable hotspot offset and the large centre-to-limb brightness temperature contrast suggest heat transport by winds moving radially away from the substellar point and towards the nightside, rather than redistributing heat to the nightside through the formation of an equatorial jet[42,43]. Lorentz forces are expected to play an important role in the atmospheric dynamics of ultra-hot Jupiters, owing to their high dayside temperatures[43–45]. Thermally ionized alkali metals coupled to an internal dynamo-driven planetary magnetic field interact with the neutral species and are expected to prevent the formation of an eastward equatorial jet[46]. By approximating the effects of the Lorentz force as a locally calculated magnetic drag force in GCMs[47] (see Methods), we find that the observed white-light curve is best explained by an internal planetary field strength of 5 G or larger, as this field strength is sufficient to prevent a discernible longitudinal shift of the hotspot from the substellar point (Fig. 4). This is further confirmed by a separate GCM considering spatially uniform drag timescales, for which we find that the case with the highest drag strength ($\tau_{drag} = 10^3$ s) produces white-light curves that best fit the observations (Fig. 4 and Extended Data Fig. 7). Furthermore, self-consistent magnetohydrodynamics (MHD) models of ultra-hot Jupiters considering the response of the magnetic field to the circulation have shown the possibility of time variability in the longitudinal hotspot offset, oscillating between the western and eastern hemispheres over timescales of 10–100 days (refs. 48,49), but further observations are needed to test this possibility.

The large wavelength coverage and high spectral and photometric precision of the NIRISS/SOSS mode of the JWST present many opportunities for the study and detailed characterization of atmospheric processes through thermal emission spectroscopy. Furthermore, planets with high signal-to-noise-ratio eclipses such as WASP-18b allow for the 3D mapping of their atmospheres to retrieve the temperature structure across the dayside as well as variations in properties such as molecular abundances[39,40]. The JWST will enable these measurements for most bright transiting exoplanets, giving rise to the possibility of studying the dynamics and chemistry of a wide range of exoplanets directly from secondary-eclipse observations.

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

¹Department of Physics, Université de Montréal, Montréal, Quebec, Canada. ²Trottier Institute for Research on Exoplanets, Université de Montréal, Montréal, Quebec, Canada. ³Department of Astronomy, University of Michigan, Ann Arbor, MI, USA. ⁴Earth and Planets Laboratory, Carnegie Institution for Science, Washington, DC, USA. ⁵School of Earth and Space Exploration, Arizona State University, Tempe, AZ, USA. ⁶Steward Observatory, University of Arizona, Tucson, AZ, USA. ⁷Department of Astronomy, Cornell University, Ithaca, NY, USA. ⁸Carl Sagan Institute, Cornell University, Ithaca, NY, USA. ⁹Department of Astronomy and Astrophysics, University of Chicago, Chicago, IL, USA. ¹⁰Department of Astronomy, University of Maryland, College Park, MD, USA. ¹¹Center for Computational Astrophysics, Flatiron Institute, New York, NY, USA. ¹²Space Telescope Science Institute, Baltimore, MD, USA. ¹³Université Côte d'Azur, Observatoire de la Côte d'Azur, CNRS, Laboratoire Lagrange, Nice, France. ¹⁴NASA Goddard Space Flight Center, Greenbelt, MD, USA. ¹⁵Tsung-Dao Lee Institute, Shanghai Jiao Tong University, Shanghai, People's Republic of China. ¹⁶School of Physics and Astronomy, Shanghai Jiao Tong University, Shanghai, People's Republic of China. ¹⁷Atmospheric, Oceanic and Planetary Physics, Department of Physics, University of Oxford, Oxford, UK. ¹⁸Department of Mathematics and Statistics, Faculty of Environment, Science and Economy, University of Exeter, Exeter, UK. ¹⁹Center for Space and Habitability, University of Bern, Bern, Switzerland. ²⁰Anton Pannekoek Institute for Astronomy, University of Amsterdam, Amsterdam, The Netherlands. ²¹Department of Physics & Astronomy, University of Kansas, Lawrence, KS, USA. ²²Department of Earth, Atmospheric and Planetary Sciences, Massachusetts Institute of Technology, Cambridge, MA, USA. ²³Kavli Institute for Astrophysics and Space Research, Massachusetts Institute of Technology, Cambridge, MA, USA. ²⁴School of Physics, University of Bristol, Bristol, UK. ²⁵Department of Earth and Planetary Sciences, University of California, Santa Cruz, Santa Cruz, CA, USA. ²⁶Department of Astronomy and Astrophysics, University of California, Santa Cruz, Santa Cruz, CA, USA. ²⁷Department of Astrophysical and Planetary Sciences, University of Colorado Boulder, Boulder, CO, USA. ²⁸European Space Agency, Space Telescope Science Institute, Baltimore, MD, USA. ²⁹Department of Physics and Astronomy, University College London, London, UK. ³⁰School of Earth and Planetary Sciences (SEPS), National Institute of Science Education and Research (NISER), Homi Bhabha National Institute (HBNI), Jatni, India. ³¹Planetary Sciences Group, Department of Physics, University of Central Florida, Orlando, FL, USA. ³²Florida Space Institute, University of Central Florida, Orlando, FL, USA. ³³Division of Geological and Planetary Sciences, California Institute of Technology, Pasadena, CA, USA. ³⁴Max Planck Institute for Astronomy, Heidelberg, Germany. ³⁵Center for Astrophysics | Harvard & Smithsonian, Cambridge, MA, USA. ³⁶Department of Earth and Planetary Sciences, Johns Hopkins University, Baltimore, MD, USA. ³⁷Department of Physics & Astronomy, Johns Hopkins University, Baltimore, MD, USA. ³⁸Johns Hopkins Applied Physics Laboratory, Laurel, MD, USA. ³⁹Indian Institute of Technology, Indore, India. ⁴⁰Centre for Exoplanets and Habitability, University of Warwick, Coventry, UK. ⁴¹Department of Physics, University of Warwick, Coventry, UK. ⁴²Bay Area Environmental Research Institute, NASA Ames Research Center, Moffett Field, CA, USA. ⁴³Department of Physics, New York University Abu Dhabi, Abu Dhabi, United Arab Emirates. ⁴⁴Center for Astro, Particle, and Planetary Physics (CAP3), New York University Abu Dhabi, Abu Dhabi, United Arab Emirates. ⁴⁵Instituto de Astrofísica, Universidad Andrés Bello, Santiago, Chile. ⁴⁶Núcleo Milenio de Formación Planetaria (NPF), Valparaíso, Chile. ⁴⁷Centro de Astrofísica y Tecnologías Afines (CATA), Santiago, Chile. ⁴⁸School of Physics and Astronomy, University of Leicester, Leicester, UK. ⁴⁹Leiden Observatory, University of Leiden, Leiden, The Netherlands. ⁵⁰INAF – Osservatorio Astrofisico di Torino, Pino Torinese, Italy. ⁵¹Space Research Institute, Austrian Academy of Sciences, Graz, Austria. ⁵²Institute of Astronomy, Department of Physics and Astronomy, KU Leuven, Leuven, Belgium. ⁵³School of Physics, Trinity College Dublin, Dublin, Ireland. ⁵⁴Universitäts-Sternwarte München, Ludwig-Maximilians-Universität München, München, Germany. ⁵⁵ARTORG Center for Biomedical Engineering Research, University of Bern, Bern, Switzerland. ⁵⁶Institute of Planetary Research (PF), German Aerospace Center (DLR), Berlin, Germany. ⁵⁷Université Paris-Saclay, Université Paris Cité, CEA, CNRS, AIM, Gif-sur-Yvette, France. ⁵⁸Laboratoire d'Astrophysique de Bordeaux, Université de Bordeaux, Pessac, France. ⁵⁹Département d'Astronomie, Université de Genève, Sauverny, Switzerland. ⁶⁰Department of Physics, Utah Valley University, Orem, UT, USA. ⁶¹Department of Physics, University of Rome "Tor Vergata", Rome, Italy. ⁶²Exzellenzcluster Origins, Garching, Germany. ⁶³Instituto de Astrofísica de Canarias (IAC), Tenerife, Spain. ⁶⁴Astronomy Department, Van Vleck Observatory, Wesleyan University, Middletown, CT, USA. ⁶⁵Department of Earth and Planetary Sciences, University of California, Riverside, Riverside, CA, USA. ⁶⁶Université Paris Cité and Université Paris-Est Creteil, CNRS, LISA, Paris, France. ✉e-mail: louis-philippe.coulombe@umontreal.ca

## Methods

### NIRISS/SOSS reduction and spectrophotometric extraction

We perform four separate reductions of the NIRISS/SOSS[13,54,55] eclipse observations of WASP-18b using the NAMELESS, nirHiss[56], supreme-SPOON[57] and transitspectroscopy[58] pipelines for inter-comparison of individual reduction steps[59]. All pipelines are built around the official STScI jwst reduction pipeline[60] with the addition of custom correction steps for systematics such as $1/f$ noise, zodiacal background and cosmic rays. Reductions are performed from either the raw uncalibrated data (NAMELESS, nirHiss and supreme-SPOON) or stage 1 (transitspectroscopy) products up to the extraction of the spectrophotometric light curves.

### NAMELESS reduction

We use the NAMELESS pipeline[59] to reduce the WASP-18b observations from the uncalibrated data products through spectral extraction. We used the jwst pipeline version 1.6.0, CRDS (Calibration Reference Data System) version 11.16.5 and CRDS context jwst_0977.pmap for the reduction. First, we go through all steps of the jwst pipeline stage 1, with the exception of the dark_current step. We skip the dark subtraction step to avoid introducing extra noise owing to the lack of a high-fidelity reference file. After the ramp-fitting step, we go through the assign_wcs, srctype and flat_field steps of the jwst pipeline stage 2; we skip the background step and apply our own custom routine for handling the background. We skip the pathloss and photom steps, as an absolute flux calibration is not needed. We perform outlier detection by computing the product of the second derivatives in the column and row directions for all frames[59]. We divide the frames into windows of $4 \times 4$ pixels, for which we then compute the local median and standard deviation of the second derivative. We flag all pixels that are $\geq 4\sigma$ away from the window median. Furthermore, we flag pixels that show null or negative counts. All flagged pixels are set equal to the local median of their window. We correct for background systematics using the following routine. First, we identified section ($x \in [5, 400], y \in [0, 20]$) as the region of the SUBSTRIP96 subarray in which the contribution from the spectral orders to the counts is minimal. Next, we compute the scaling factor between the median frame and the model background provided on the STScI JDox User Documentation[61] within the aforementioned region. We consider the 16th percentile of the distribution as the scaling value and subtract the scaled background from all integrations. We pay close attention to the $1/f$ correction for these observations, as the magnitude of the spectral trace variation is highly dependent on wavelength in the secondary eclipse. Therefore, we consider all columns independently when scaling the trace to compute the $1/f$ noise. Furthermore, we treat this correction in two parts, as we observe a tilt event[20,21] around the 1,336th integration (Extended Data Fig. 2), possibly because of a sudden movement in one of the segments of the primary mirror, resulting in a change in the morphology of the trace that manifests as a sudden decrease in flux. First, we compute the median columns $\bar{c}$ before and after the tilt event; we use integrations 300–900 and 1,350–1,900. Then, we define a given column $j$ and row $k$ at an integration $i$ as the sum of the scaled median column $m_{i,j}\bar{c}_{j,k}$ and the $1/f$ noise $n_{i,j} = c_{i,j,k} - m_{i,j}\bar{c}_{j,k}$. Using the errors $\varepsilon_{i,j,k}$ returned by the jwst pipeline, we solve for the values of $m_{i,j}$ and $n_{i,j}$ that minimize the chi-square between the observed and the scaled columns $\chi^2 = \sum_k [(c_{i,j,k} - m_{i,j}\bar{c}_{j,k} - n_{i,j})/\varepsilon_{i,j,k}]^2$. These values of the scaling $m_{i,j}$ and $1/f$ noise $n_{i,j}$ are obtained by imposing $\partial\chi^2/\partial m_{i,j}$, $\partial\chi^2/\partial n_{i,j} = 0$, such that

$$m_{i,j} = \frac{\sum_k \frac{\bar{c}_{j,k}}{\varepsilon_{i,j,k}^2}\left(\sum_k \frac{1}{\varepsilon_{i,j,k}^2}\right)^{-1}\sum_k \frac{c_{i,j,k}}{\varepsilon_{i,j,k}^2} - \sum_k \frac{c_{i,j,k}\bar{c}_{j,k}}{\varepsilon_{i,j,k}^2}}{\left(\sum_k \frac{\bar{c}_{j,k}}{\varepsilon_{i,j,k}^2}\right)^2\left(\sum_k \frac{1}{\varepsilon_{i,j,k}^2}\right)^{-1} - \sum_k \left[\frac{\bar{c}_{j,k}}{\varepsilon_{i,j,k}}\right]^2} \quad (1)$$

and

$$n_{i,j} = \sum_k \frac{c_{i,j,k}}{\varepsilon_{i,j,k}^2}\left(\sum_k \frac{1}{\varepsilon_{i,j,k}^2}\right)^{-1} - m_{i,j}\sum_k \frac{\bar{c}_{j,k}}{\varepsilon_{i,j,k}^2}\left(\sum_k \frac{1}{\varepsilon_{i,j,k}^2}\right)^{-1}. \quad (2)$$

We then subtract the measured values of $n_{i,j}$ from all columns and integrations. We set the error $\varepsilon_{i,j,k}$ to $\infty$ (in which $\infty$ here is defined as the IEEE 754 floating-point representation of positive infinity) for all pixels that have non-zero data quality flags returned by the jwst pipeline such that they are not considered in the fit of the $1/f$ noise. We also set the errors to $\infty$ in the region $x \in [76, 96], y \in [530, 1,350]$ of the detector, in which a portion of the second spectral order is visible. This is appropriate treatment as the $1/f$ noise scales independently across each order owing to the difference in wavelength coverage. After correction of the $1/f$ noise, we trace the location of order 1 on the detector by computing the maximum of the trace convolved with a Gaussian filter for all columns. We further smooth the positions of the trace centroids using a spline function. Finally, we perform a box spectral extraction of the first order using the transitspectroscopy.spectroscopy.getSimpleSpectrum routine with an aperture diameter of 30 pixels.

### nirHiss reduction

We use the nirHiss Python open-source data-reduction pipeline as described in ref. 59. To summarize, this pipeline uses Eureka![62] to go from the stage 0 JWST outputs to stage 2 calibration, which applies detector-level corrections, produces count-rate images and calibrates individual exposures. From the stage 2 'calints' FITS files, we use nirHiss to correct for background sources, $1/f$ noise, cosmic-ray removal, trace identification and spectral extraction. We follow two steps for trace identification. First, we use the $(x, y)$ position of the trace from the 'jwst_niriss_spectrace_0022.fits' reference file, with $y$ values offset by about 25 pixels, to identify order 2. We mask this region, such that it does not contaminate the trace identification or background routine later on. Next, we use the nirHiss.tracing.mask_method_edges function. This technique identifies the edges of order 1 using a canny edge-detection method from scikit-image, an open-source image-processing package[63]. This method uses the derivative of a Gaussian function to identify regions with the maximum gradient. From this step, the potential edges are narrowed down to 1-pixel curves along the maxima. This results in an image in which the outline of order 1 is presented. We identify the median location along the column from the top and bottom edges of order 1 and smooth the trace by fitting a fourth-order polynomial. We find that a fourth-order polynomial best fits the overall shape of both orders, whereas a sixth-order polynomial overfits and a second-order polynomial underfits the order profile. We use the trace to mask the location of order 1 when stepping through the nirHiss background routines. For background treatment, we follow a similar method presented in ref. 59, namely, we identify a region without substantial contamination from the spectral trace and scale this region to the same region on the model background on the STScI JDox User Documentation. We use the region $x \in [4, 250], y \in [0, 30]$ and find an average scaling factor of 0.6007. We apply this scaling factor to the model background and subtract it from all integrations. Next, we remove $1/f$ noise in a similar manner to transitspectroscopy and scale this $1/f$ noise treatment to the out-of-eclipse integrations (0–1,250 and 1,900–2,500). We remove cosmic rays using the L.A.Cosmic technique[64]. Finally, we extract the spectra using the optimal extraction routine, which is a robust means to simultaneously remove further bad pixels/cosmic-ray events while placing non-uniform weighting on each pixel to negate distortion produced by the spatial profile[65]. We use a normalized median image to best capture the unique NIRISS/SOSS spatial profile.

### supreme-SPOON reduction

We follow a similar approach with supreme-SPOON as presented in ref. 59. We start from the raw uncalibrated data files, which we

downloaded from the Mikulski Archive for Space Telescopes (MAST) archive, and process them through the supreme-SPOON stage 1, which performs the detector-level calibrations, including superbias subtraction, saturation flagging, jump detection and ramp fitting. As with the previous pipelines, we do not perform any dark current subtraction. supreme-SPOON further treats the 1/f noise at the group level. This is done by subtracting a median stack of all in-eclipse integrations, scaled to the flux level of each individual integration through the white-light curve, to create a difference image revealing the characteristic 1/f striping. A column-wise median of the $n$th difference image is then subtracted from the corresponding integration. supreme-SPOON also removes the zodiacal background signal directly before the 1/f correction step by scaling the SUBSTRIP96 SOSS background model provided by the STScI to the flux level of each integration, as described in ref. 59. We then pass the stage 1-processed files through supreme-SPOON stage 2, which performs further calibrations such as flat fielding and hot-pixel interpolation. We extract the stellar spectra at the pixel level using a simple box aperture extraction with a width of 30 pixels centred on the order 1 trace, as the dilution resulting from the order overlap with the second order has been shown to be negligible[66,67]. The $y$-pixel positions of the trace are determined through the edgetrigger algorithm[67]. We find that the extracted trace positions match those measured during commissioning and included in the spectrace reference file and we therefore use the default JWST wavelength solution.

## transit spectroscopy reduction

We follow a similar approach adopted by the transit spectroscopy pipeline discussed in ref. 59. This reduction starts from the _rateints.fits files that were processed by the jwst pipeline from STScI. We scaled the zodiacal background model provided on the STScI JDox User Documentation to the observed 2D spectra in the box delimited by pixels $x \in [10, 250]$, $y \in [10, 30]$. The scaled background is then subtracted from each integration. In summary, the procedure to remove the 1/f noise is as follows: we take the median of all integrations and subtract it from each integration, which leaves predominantly the 1/f noise. We then take the column-by-column median of this residual noise, considering only the pixels that are 20 pixels away from the centre of the trace, and assume it is representative of the structure of the 1/f noise of the images. These values are then subtracted from each column. For the spectral extraction, we used the transitspectroscopy.spectroscopy. getSimpleSpectrum routine with an aperture width of 30 pixels. We removed the outliers of the extracted spectra caused by cosmic rays or deviating pixels by taking the combined median of all spectra and flagging outlier points that deviate by more than $5\sigma$ from this median spectrum. The flagged wavelength bins are then corrected by taking the mean of the neighbouring bins.

## Spectrophotometric-light-curve creation

From the aforementioned stellar spectral extraction pipelines, we create and fit models to the spectrophotometric light curves $F(t, \lambda)$. The light curves are composed of three distinct signals: the planetary flux throughout partial phase curve and eclipse $F_p$, the stellar flux $F_*$ and the systematics $S$, which we model as a function of time $t$ and wavelength $\lambda$ by means of equation (3).

$$F(t, \lambda) = S(t, \lambda) \left[ \frac{F_p(t, \lambda, \theta)}{F_*(t, \lambda)} + 1 \right] \qquad (3)$$

As the main scientific quantity of interest is the planetary signal $F_p(t, \lambda, \theta)$, in which $\theta$ represents the planetary orbital parameters, we aim to properly characterize and correct for the stellar flux and systematics. Light-curve fitting is performed in two separate steps: (1) we fit the white-light curve and (2) we run individual fitting for each spectrophotometric bin. The values of the orbital parameters obtained from the white-light curve are fixed in the spectrophotometric light curves. The following sections describe our treatment of the planetary flux $F_p(t, \lambda, \theta)$, the stellar variability $F_*(t, \lambda)$ and the systematics $S(t, \lambda)$.

## Light-curve component 1: planetary flux

Despite the fact that the main target of these observations was the secondary eclipse of WASP-18 b, we also capture a portion of its phase curve during the before-eclipse and after-eclipse baseline. Over the course of the observations, the planet rotates 107°, revealing substantial information on the spatial distribution of its atmosphere. To model the planetary flux in time, we consider a second-order harmonic function

$$F_p(t, \lambda, \theta) = f(t, \theta) \sum_{n=1}^{2} \left[ F_n + F_n \cos\left( \frac{2\pi n}{P} [t - t_n(\lambda)] \right) \right], \qquad (4)$$

in which $0 \le f(t, \theta) \le 1$ is the time-dependent, visible fraction of the planetary projected disk as a function of the orbital parameters $\theta$ and $P$ is the orbital period. The harmonics consist of a term describing the semiamplitude $F_n$ of the planetary flux variation, as well as the time $t_n$ at which the harmonic reaches its maximum. The visible fraction is computed using the normalized secondary-eclipse light curve modelled with the batman Python package[68]. The second-order harmonic function provides sufficient precision for the noise floor of the JWST[69]. We fit for the orbital parameters that dictate the shape and duration of the eclipse: the time of superior conjunction $T_{sec}$ (U[2459802.78, 2459802.98]), the impact parameter $b$ (N[0.360, 0.026$^2$]), as well as the semi-major axis to stellar radius ratio $a/R_*$ (N[3.496, 0.029$^2$]). The normal priors considered for the semi-major axis and impact parameter use the median and $1\sigma$ uncertainties of the constraints obtained from TESS (see Extended Data Table 1) as the centre and standard deviation of the priors. Those priors are used because of the precise constraints obtained from several TESS transits and they are free from correlations with a potentially non-uniform dayside, as opposed to our secondary-eclipse light curve. We opt to keep $a/R_*$ and $b$ free rather than fixing them to the TESS values to ensure that the other parameters retrieved from the white-light-curve fit are marginalized over the TESS uncertainties. We assume the orbit to be circular, which is justified by the TESS analysis, as it finds a strong Bayesian information criterion[70] (BIC) and Akaike information criterion[71] (AIC) preference for the non-eccentric orbit. Given the close proximity of WASP-18b to its host star, strong tidal interactions lead to the ellipsoidal deformation of the planet and its host. Past studies have shown that this deformation for WASP-18b is around $2.5 \times 10^{-3}\,R_p$, leading to a variation of the flux of order unity ppm, and is thus negligible[16,72]. Finally, near the lower end of the first order (0.85 μm), there is also a contribution to the observed flux from the stellar light reflected by WASP-18b. However, the upper limits of the geometric albedo ($A_g < 0.048$ (ref. 73) and $A_g = 0.025 \pm 0.027$ (ref. 74)) obtained from TESS correspond to a reflected-light contribution of <35 ppm near 0.8 μm. We therefore do not consider a specific term for the reflected light and instead assume that this will be fit by the second-order harmonic function.

## Light-curve component 2: stellar variability

We consider three phenomena that can lead to temporal changes in the observed stellar flux: stellar activity, $A$, ellipsoidal variations, $E$, and Doppler boosting, $D$. Stellar activity, generally caused by the presence and movement of star spots on the stellar hemisphere visible to the observer, leads to variations in the observed stellar spectrum on a timescale that is on the order of the stellar rotation period. Past observations of WASP-18 have constrained this period to be $P_{rot} \approx 5.5$ days (ref. 75). Despite this relatively short period with respect to stars of similar physical properties (for example, the effective temperature, $T_{eff}$, and luminosity, $L_\odot$)[76], the star shows abnormally low activity in the ultraviolet and X-ray domains, possibly because of tidal interactions with WASP-18b disrupting its dynamo[77-79]. As we expect the rotational modulation to be on a relatively long timescale compared with our

observations and its amplitude to be low, we do not directly fit this term and instead assume it to be handled by the systematics model. For the ellipsoidal variation and Doppler boosting, they are both caused by the influence of WASP-18b on its host star. Although the ellipsoidal deformation of WASP-18b leads to a negligible impact on the phase curve, the same is not true for its host. The stellar ellipsoidal effect, with maxima fixed at quadrature when the projected area is at its highest, is found to be of semiamplitude 172.2 ppm from the TESS analysis. Following previous analyses of HST observations[16], we consider ellipsoidal variation to be achromatic and fix its amplitude to the TESS value for the full NIRISS/SOSS wavelength range. The Doppler boosting effect is a result of the Doppler shift of the stellar spectral energy distribution as its radial velocity varies throughout its orbit. We fix this amplitude to 21.8 ppm, with the maximum at phase 0.25, as carried out in ref. 16. The observed stellar flux is therefore described as the sum of the ellipsoidal variation and Doppler boosting to the mean stellar flux in time $\overline{F}_*(\lambda)$.

$$F_*(t,\lambda) = \overline{F}_*(\lambda) - D\sin\left(\frac{2\pi}{P}[t - T_{\mathrm{sec}}]\right) - E\cos\left(\frac{4\pi}{P}[t - T_{\mathrm{sec}}]\right) \qquad (5)$$

### Light-curve component 3: systematics model
The white and spectrophotometric light curves show two distinct important systematics: a sudden drop in flux caused by a tilt event shortly after the beginning of full eclipse, as well as high-frequency variations in the flux throughout the observations caused by small changes in the morphology of the trace. We track the trend of these systematics throughout the observations by performing incremental principal components analysis (PCA) with the open-source scikit-learn[80] package on the processed detector images (Extended Data Fig. 2). The first principal component is the tilt event, which we use to determine the exact integration in which it occurs. We handle the tilt event in the white and spectrophotometric light curves by fitting for an offset in flux for the data after the 1,336th integration. We also observe two principal components analogous to the $y$ position and full width at half maximum (FWHM) of the trace in time. We find that these two components are correlated to short-frequency variations in the light curves and therefore detrend linearly against them at the fitting stage. We find that, despite having a lower variance than the $y$ position, the variation of the FWHM has a larger effect on the light curves when using box extraction. Finally, we fit for a linear trend in time to account for any further potential stellar activity and instrumental effect.

We note that considering a second-order polynomial trend or higher in time for the systematics results in notable correlation with the partial-phase-curve information. However, a linear-trend systematics model has been found sufficient to fit previous NIRISS/SOSS observations[59,81]. Furthermore, we find that the curvature around the secondary eclipse increases monotonically with wavelength, which is expected from the planetary signal and inconsistent with stellar activity as well as instrumental effects.

### Light-curve fitting
Light-curve fitting is performed on the extracted spectrophotometric observations using the ExoTEP framework[22]. With the orbital parameter values constrained from the white-light curve (see Extended Data Table 1), we solely fit for the planetary flux and systematics for each spectrophotometric light curve. The retrieved values of $a/R_*$ and $b$ are within $1\sigma$ of the TESS values and such deviations do not affect the retrieved thermal emission spectrum as it is insensitive to the orbital parameters compared with transmission spectroscopy. We further explore the impact of our uncertainties on the retrieved map in the 'Secondary-eclipse mapping' section and find it to be robust against variations of the orbital parameters. We chose a resolution of 5 pixels per bin for our spectrum, corresponding to 408 spectrophotometric bins, to mitigate potential correlation between wavelengths in the

atmospheric retrievals, as pixels in the spectral direction are not independent. All fits for the 408 bins are then done independently to obtain the planetary flux at secondary eclipse for all bins. Light-curve fits are performed using the affine-invariant Markov chain Monte Carlo (MCMC) ensemble sampler emcee[82], using 20,000 steps and four walkers per free parameter. The first 12,000 steps, 60% of the total amount, are discarded as burn-in to ensure that the samples are taken after the walkers have converged. The samples from the white and spectrophotometric light curves are used to produce two science products: the detrended white-light curve and dayside thermal emission spectrum. The detrended white-light curve is obtained by dividing out the best-fit systematics model and subtracting the stellar variability from the light curve to isolate the planetary signal. For the dayside thermal emission, the median values and uncertainties of $F_p(T_{\mathrm{sec}}, \lambda)$ are computed from the samples of the parameters of equation (4).

### TESS phase-curve analysis
During the TESS Primary Mission, the WASP-18 system was observed in sectors 2 and 3 (22 August to 18 October 2018). The full-orbit phase curve was analysed in several previous publications[73,83], which reported a robust detection of the planet's secondary eclipse and high signal-to-noise measurements of the planet's phase-curve variation and signals corresponding to the ellipsoidal distortion and Doppler boosting of the host star. During the continuing TESS Extended Mission, the spacecraft reobserved WASP-18 in sectors 29 and 30 (26 August to 21 October 2020). We carried out a follow-up phase-curve analysis of all four sectors' worth of TESS data, following the same methods used previously.

The data from the TESS observations were processed by the Science Processing Operations Center (SPOC) pipeline, which yielded near-continuous light curves at a 2-min cadence. As well as the raw extracted flux measurements contained in the simple aperture photometry light curves, the SPOC pipeline also produced the pre-search conditioning light curves, which have been corrected for common-mode instrumental systematics trends that are shared among all sources on the corresponding detector. We used these pre-search conditioning light curves for our phase-curve analysis. After dividing the light curves into individual segments that are separated by the scheduled momentum dumps of the spacecraft, we fit each segment to a combined phase curve and systematics model. The astrophysical phase-curve model consists of two components describing the planetary and stellar fluxes:

$$F_p(t) = \overline{f}_p - F_{\mathrm{atm}}\cos(\phi) \qquad (6)$$

$$F_*(t) = 1 + D\sin(\phi) - E\cos(2\phi) \qquad (7)$$

The Doppler boosting $D$ and ellipsoidal distortion $E$ semiamplitudes are defined as before. Here the orbital phase of the planet is defined relative to the mid-transit time $T_0$: $\phi = 2\pi(t - T_0)$. The phase-curve contribution of the planet has a single mode with a semiamplitude of $F_{\mathrm{atm}}$ and oscillates around the average relative planetary flux $\overline{f}_p$. The transit and secondary-eclipse light curves $\phi_t$ and $\phi_e$ were modelled using batman with quadratic limb darkening. In this parametrization, the secondary-eclipse depth (that is, dayside flux) and nightside flux are $\overline{f}_p + F_{\mathrm{atm}}$ and $\overline{f}_p - F_{\mathrm{atm}}$, respectively. For the systematics model, we used polynomials in time and chose the optimal polynomial order for each segment individually that minimized the BIC.

We used ExoTEP to calculate the posterior distributions of the free parameters through a joint MCMC fit of all light-curve segments. To reduce the number of free parameters in the joint fit, we first carried out fits to smaller groups of light-curve segments corresponding to each TESS sector and then divided the light-curve segments by the best-fit systematics model. In the final joint fit of the systematics-corrected TESS light curve, no further systematics parameters were included. We accounted for time-correlated noise (that is, red noise)

by fixing the uncertainty of all data points within each sector to the standard deviation of the residuals, multiplied by the fractional enhancement of the average binned residual scatter from the expected Poisson noise scaling across bin sizes ranging from 30 min to 8 h (ref. 83). As well as the phase-curve parameters described above ($\bar{f}_p$, $F_{atm}$, $D$ and $E$), we allowed the mid-transit time $T_0$, orbital period $P$, relative planetary radius $R_p/R_*$, scaled orbital semi-major axis $a/R_*$, impact parameter $b$ and quadratic limb-darkening coefficients $u_1$ and $u_2$ to vary freely.

The results of our TESS phase-curve fit are listed in Extended Data Table 1. The revised values for the orbital ephemeris and transit shape parameters are statistically consistent with the published results from the previous TESS phase-curve analyses[73,83], while being substantially more precise. We used the median and $1\sigma$ uncertainties of $a/R_*$ and $b$ as Gaussian priors in the NIRISS/SOSS white-light-curve fit. We also used the median values of $P$, $R_p/R_*$, $D$ and $E$ as fixed parameters for the NIRISS/SOSS white and spectrophotometric light curves fit. We obtained a secondary-eclipse depth of $357 \pm 14$ ppm and a nightside flux that is consistent with zero. All three phase-curve amplitudes were measured at high signal-to-noise ratio: $F_{atm} = 177.5 \pm 5.7$ ppm, $D = 21.8 \pm 5.2$ ppm and $E = 172.2 \pm 5.6$ ppm.

## Secondary-eclipse mapping

To perform eclipse mapping, we use both ThERESA[40] and the methods in ref. 39, which are separate implementations of the same process introduced in ref. 38 when applied to white-light curves, to cross-check our results. First, we generate a basis set of light curves from spherical harmonic maps[84,85] with degree $l_{max}$ and then transform these light curves to a new, orthogonal basis set of 'eigencurves' using PCA. Each eigencurve corresponds to an 'eigenmap', the planetary flux map that, when integrated over the visible hemisphere at each exposure time, generates the corresponding eigencurve.

We then fit the white-light curve with a linear combination of a uniform-map light curve, the $N$ most informative (largest eigenvalue) eigencurves and a constant offset term to adjust for the fact that the observed planetary flux during eclipse (when the planet is entirely blocked by the star) must be equal to 0 and to allow for adjustments to light-curve normalization. Because the eigenmaps represent differences from a uniform map, it is possible to recover a fit that contains regions of negative planet emission. This is physically impossible, so we impose a positivity constraint on the total flux map. Although the eigenmaps are mathematically defined across the entire planetary sphere, our observations only constrain the portion of the planet that is visible during the observation, so we only enforce the flux-positivity condition in the visible region of the planet. Although this positivity condition could introduce a Lucy–Sweeney[86] bias near zero flux, we note that our fitted maps (and GCM predictions) are far from negative, and increasing this boundary to, for example, 300 K leads to no change in the results. We test all combinations of $l_{max} \le 6$ and $N \le 8$ using a least-squares minimization and select the optimal values by minimizing the BIC.

We find that the fit with the lowest BIC to the broadband light curve has $l_{max} = 5$ and $N = 5$. However, the fit with $l_{max} = 2$ and $N = 5$ was only slightly less preferred, so here we explore the inferred brightness distribution from both solutions. Extended Data Fig. 8 shows the resulting light curve for the $l_{max} = 5$, $N = 5$ solution after sequential subtraction of each eigencurve. The preference for a fit with five eigencurves is driven by the residuals in ingress and egress, which can be seen by eye and are not sufficiently corrected for with a uniform map. Including the uniform-map light curve and the constant term, the fit thus contained a total of seven free parameters. We used a MCMC procedure to estimate parameter uncertainties. For the analysis following ref. 39, we test for convergence of the MCMC by ensuring that the chain length is 50 times the autocorrelation timescale, whereas for the analysis using ThERESA[40], we use the Gelman–Rubin convergence test[87] and achieve values ≤1.00006.

The resulting weights of each eigencurve are then applied to the corresponding eigenmaps to generate a flux map of the planet. We convert the star-normalized flux map to brightness temperature by assuming that the planet is a blackbody and the star emits as a PHOENIX[41] spectrum calculated with PyMSG[88], both integrated over the NIRISS/SOSS throughput. We estimate temperature-map uncertainties by computing a subsample of maps from the MCMC posterior distribution and calculating 68.3%, 95.5% and 99.7% quantiles at each location, including the effects of uncertainties in planetary radius, stellar temperature and stellar log $g$.

Figure 4 shows the resulting broadband brightness temperature map for the $l_{max} = 5$, $N = 5$ case and longitudinal brightness temperature profiles for both the $l_{max} = 5$, $N = 5$ and $l_{max} = 2$, $N = 5$ cases, calculated by averaging meridian flux at each longitude weighted by $\cos(\text{latitude})^2$. Furthermore, we compare the equatorial slices to predictions from several GCMs (see 'GCMs' section). We note that not all structures on a planetary map will leave an observable signature in a secondary-eclipse light curve. When comparing GCMs to secondary-eclipse maps, it is important to only compare the components of GCM maps that can be physically accessed with eclipse mapping. Therefore, we use the methods in ref. 89 to separate each GCM map into the 'null space', or components that are inaccessible to eclipse-mapping observations, and the 'preimage', or components that are accessible through mapping. Figure 4 compares the longitudinal temperature trends from only the preimage of each GCM to the observed map. We find that both map solutions agree on a steep gradient in temperature near the limbs, which is well matched by GCM predictions. Furthermore, both maps show a temperature distribution roughly symmetrical in longitude about the substellar point. However, the two maps disagree on the exact shape of the brightness distribution. The $l_{max} = 5$, $N = 5$ map shows an extended hot-plateau region of roughly constant temperature from −40° to +40° in longitude, whereas the $l_{max} = 2$, $N = 5$ map shows a more concentrated hotspot with a steady decrease in temperature away from the substellar point. As these maps both fit the data with similar BIC, the present data do not give us the necessary precision to determine which solution represents the true temperature distribution of WASP-18b. Future observations at higher precision may distinguish between these two modes of solutions.

To test our ability to constrain latitudinal structures, we performed two eclipse-mapping fits: the eigenmapping fit presented above and a fit in which the initial basis set of maps is a longitudinal Fourier series that is constant with latitude (see Extended Data Fig. 9). Both methods retrieve similar longitudinal temperature structures, with steep gradients near the limb and an extended hot plateau. However, the constant-with-latitude map is also able to fit the data well, indicating a lack of constraints on latitudinal features within the uncertainties on the data. This is not unexpected, as the relatively low impact parameter ($b = 0.34$) of WASP-18b results in a lower amount of latitudinal information contained in the secondary-eclipse signal. Further observations of WASP-18b, or of planets with higher impact parameter, may enable us to pull latitudinal signals out of the noise.

Our eclipse mapping assumes that the orbital parameters of the system are precisely known relative to data uncertainties, a safe assumption with Spitzer data[38]. With the JWST, data quality may be high enough that uncertainties on orbital parameters impart substantial uncertainty on the mapping results. To test this, we ran analyses with impact parameter, orbital semi-major axis and eclipse time fixed to values $\pm 1\sigma$. In some cases, this led to a 'hotspot' model such as the red one in Fig. 4 being preferred over a 'plateau' model such as the blue one, which is unsurprising given their similar statistical preference. However, all resulting maps were well within the uncertainties of one of the two models presented. We note that, although the eccentricity is kept fixed to zero throughout this analysis, as justified by the preference for a circular orbit in the TESS analysis, considering an eccentric orbit would allow to the first order for variations in mid-eclipse time and

eclipse duration[90]. Past photometric and radial-velocity observations of WASP-18b have found a small but non-zero eccentricity for WASP-18b on the order of $e = 0.008$ (refs. 91–93), corresponding to an offset of the time of mid-eclipse of 9 s, as well as a difference of 120 s between the transit and eclipse durations. These differences in eclipse timing and duration are of the same magnitude as those induced when varying $T_{sec}$, $a/R_*$ and $b$ (8 s for the mid-eclipse time and 90 s for the eclipse duration). Therefore, performing the mapping considering a circular orbit while varying $T_{sec}$, $a/R_*$ and $b$ is analogous to effects that could be expected from an eccentric orbit.

## Atmospheric retrieval

We perform 1D atmospheric retrievals on the NAMELESS reduction at a resolution of 5 pixels per bin (408 bins) using four techniques with varying levels of physical assumptions: a self-consistent radiative–convective–thermochemical equilibrium grid retrieval (ScCHIMERA), a chemical equilibrium with free temperature–pressure profile retrieval (SCARLET), a free-chemistry retrieval with thermal dissociation (HyDRA) and a free-chemistry retrieval with abundances assumed constant with altitude (POSEIDON). None of the retrievals used here consider the presence of clouds, as the dayside of giant exoplanets are expected to be cloudless at the equilibrium temperature $T_{eq} = 2{,}429$ K (ref. 24) of WASP-18b (ref. 94). All retrieval methods considered the same PHOENIX stellar spectrum[23], produced using previously published parameters for WASP-18 (that is, $T_{eff} = 6{,}435$ K, $\log g = 4.35$ and [Fe/H] = 0.1 (refs. 24,95)), to convert from model planet flux spectra to $F_p/F_s$ values. We chose to use a model stellar spectrum instead of the extracted spectrum to avoid the possible introduction of systematic errors through the process of absolute flux calibration.

## 1D-RCTE grid retrieval

We use a 1D radiative–convective–thermochemical equilibrium (1D-RCTE) grid-retrieval-based method, ScCHIMERA[7,96], with the opacity sources described in refs. 2,97. These 1D-RCTE models assume cloud-free 1D-RCTE using the methods described in ref. 98 to solve for the net flux divergence across each layer of the atmosphere and the Newton–Raphson iteration scheme in ref. 99 to march towards an equilibrium vertical temperature structure. The NASA Chemical Equilibrium with Applications 2 (CEA2) routine[100] is used to compute the thermochemical equilibrium gas and condensate mole fractions for hundreds of relevant species. Opacities are computed with the correlated-$k$ random-overlap-resort-rebin method[101]. Input elemental abundances from ref. 102 are scaled to a given metallicity ([M/H]) and carbon-to-oxygen (C/O) ratio.

Using the planetary and stellar parameters of WASP-18b, we produced a grid of 2,730 1D-RCTE models and resulting top-of-atmosphere thermal emission spectra spanning the atmospheric C/O (0.01–2.0), [M/H] (−2.0–2.0, for which 0 is solar, 1 is 10× etc.) and heat redistribution ($f$, 1.0–2.8, in which 1 = full, 2 = dayside and 2.67 is the maximum value, as defined in ref. 16). We also include a scale factor, $A_{HS}$ (allowed to vary from 0.5 to 2.0), that multiplies the planetary flux by a constant to account for a hotspot area fraction emitting most of the observed flux[51]. The PyMultiNest[103] routine is used to sample the 1D-RCTE spectra through interpolation (and subsequent binning to the data wavelength bins) to obtain posterior probability constraints on the above parameters. We have made public our grid models, including temperature–pressure profiles, molecular abundances and emission spectra, as well as extra figures showing the posteriors of retrieved parameters and the impact of each parameter on the spectrum. This can be found on Zenodo at https://doi.org/10.5281/zenodo.7332105.

## Chemical-equilibrium and free-temperature retrieval

We use the SCARLET atmospheric-retrieval framework[22,104–106] to perform a chemical-equilibrium retrieval with a free temperature–pressure profile on our retrieved dayside thermal emission spectrum.

The SCARLET forward model computes the emergent disk-integrated thermal emission for a given set of molecular abundances, temperature structure and cloud properties. The forward model is then coupled to the affine-invariant MCMC ensemble sampler emcee[82] to constrain the atmospheric properties. Owing to the high dayside temperature and large pressures examined through thermal emission spectroscopy, we assume that the atmosphere is in thermochemical equilibrium. For the equilibrium chemistry, we consider the following species: $H_2$, H, $H^-$ (refs. 107,108), He, Na, K (refs. 109,110), Fe, $H_2O$ (ref. 111), OH (ref. 112), CO (ref. 112), $CO_2$ (ref. 112), $CH_4$ (ref. 113), $NH_3$ (ref. 114), HCN (ref. 115), TiO (ref. 116), VO (ref. 117) and FeH (ref. 118). The abundances of these species are interpolated in temperature and pressure using a grid of chemical-equilibrium abundances from FastChem 2 (ref. 119), which includes the effects of thermal dissociation for all the species included in the model. These abundances also vary with the atmospheric metallicity, [M/H] (U[−3, 3]), and carbon-to-oxygen ratio, C/O (U[0, 1]), which are considered as free parameters in the retrieval. For the temperature structure, we use a free parametrization[120] that here fits for $N = 10$ temperature points (U[100, 4,500] K) with fixed spacing in log-pressure ($P = 10^2$–$10^{-6}$ bar). Although this parametrization is free, it is regularized by a prior punishing for the second derivative of the profile using a physical hyperparameter, $\sigma_s$, with units of kelvin per pressure decade squared (K dex$^{-2}$). This prior is implemented to prevent overfitting and non-physical temperature oscillations at short pressure-scale lengths. For this work, we use a hyperparameter value of $\sigma_s = 1{,}000$ K dex$^{-2}$, corresponding to a low punishment against second derivatives, as we want the retrieval to explore freely the temperature–pressure profile parameter space. We note that further lowering this punishment does not affect the retrieved temperature–pressure profile. Finally, we fit for an area fraction $A_{HS}$ (U[0, 1]) that is multiplied directly with the thermal emission spectrum, for a total of 14 free parameters. This factor is used to compensate for the presence of a hotspot that, although taking up only a portion of the planetary disk, contributes almost completely to the observed emission[51]. For the retrieval, we use four walkers per free parameter and consider the standard chi-square likelihood for the spectrum fit. We run the retrieval for 25,000 steps and discard the first 15,000 steps, 60% of the total amount, to ensure that the samples are taken after the walkers have converged. Spectra are initially computed using opacity sampling at a resolving power of $R = 15{,}625$, which is sufficient to simulate JWST observations[121], convolved to the instrument resolution and subsequently binned to the retrieved wavelength bins.

Owing to the large mass of WASP-18b of $M_p = 10.4$ $M_J$, an important amount of rocky and icy material can be accreted without markedly changing the overall metallicity of the planet. As a zeroth-order estimate, we assume that the planet formed with exactly solar metallicity. Then, the mass of metals accreted needed to increase the overall metallicity to $N$ times solar is given by $Z_\odot M_p(N − 1)$, in which $Z_\odot = 0.0134$ (ref. 25) is the solar metal mass fraction. From this relation, and assuming that the envelope of the planet is well mixed, we relate our retrieved metallicity probability posterior to the mass of metals accreted.

We quantify the impact of the stellar spectrum considered for the analysis on the retrieved atmospheric properties by running the same retrieval while varying the stellar spectrum. First, we explore the impact of the PHOENIX stellar-model parameters by varying them within their $1\sigma$ uncertainties (50 K for $T_{eff}$, 0.05 for $\log g$ and 0.1 for [M/H]). We find these variations to have minimal impact on the retrieved metallicity with the measured median varying at most by 0.04 dex (about $0.15\sigma$). The same is true for the C/O upper limit, with all retrieved upper limits being within 0.06 of the retrieval with the standard stellar parameters, with the exception of the $T_{eff} = 6{,}385$ K case, which retrieves C/O < 0.64 at $3\sigma$ but does not affect our conclusions on the formation and migration history of WASP-18b. Second, we test the impact of the type of stellar model considered by also running the retrieval with an ATLAS9 stellar model[122] ($T_{eff} = 6{,}435$ K, $\log g = 4.35$, [M/H] = 0.1). We find that the effect on the results are similar to those observed when varying the

stellar parameters within their $1\sigma$ uncertainty, with a retrieved metallicity measurement of $[M/H] = 0.05^{+0.26}_{-0.25}$ and C/O $3\sigma$ upper limit of 0.603. Finally, we test the effect of using the flux-calibrated spectrum on the retrieved atmospheric properties. The use of a flux-calibrated spectrum, measured directly from the NIRISS/SOSS observations, was avoided owing to some slight issues found in the CRDS reference files available at present used in the photom step of the jwst pipeline stage 2. The most recent reference file, photom_0034, is able to reproduce accurately the continuum from the PHOENIX model considered in the main retrieval but shows substantial noise in the observed spectrum. We also looked at reference file photom_0037, which was produced from ground data and does not account for the larger-than-expected throughput that was observed on sky[20]. Despite this, we perform a retrieval on the flux-calibrated stellar spectrum obtained by smoothing the response curve of reference file photom_0034 with a median filter of width 100. The retrieval ran on the flux-calibrated stellar spectrum retrieves a metallicity $[M/H]$ of $0.11^{+0.24}_{-0.68}$ and a C/O $3\sigma$ upper limit of 0.739.

We also quantify the impact of the choice of reduction on the retrieved atmospheric properties by performing the same retrieval on the four spectra shown in Extended Data Fig. 3. We find that all reductions retrieve metallicities that are within $1\sigma$ of the NAMELESS reduction, with $[M/H]$ values of $0.00^{+0.38}_{-0.66}$, $0.05^{+0.30}_{-0.33}$ and $0.37^{+0.38}_{-0.31}$ for the nirHiss, transitspectroscopy and supreme-SPOON reductions, respectively. We also retrieve C/O $3\sigma$ upper limits of 0.749, 0.602 and 0.627 in that same order. We note that the slightly higher metallicity retrieved from the supreme-SPOON reduction is most probably because of the downward slope longwards of 2 μm, possibly caused by dilution of the signal through the process of background subtraction or $1/f$ correction, which is not observed in the nirHiss and transitspectroscopy reductions. We also find that the nirHiss retrieves larger uncertainties on the measured $[M/H]$ and C/O, caused by the larger scatter at short wavelengths, which is possibly introduced through the optimal extraction process, as this effect is not seen in the reductions using box extraction.

## Free-chemistry and free temperature–pressure profile retrieval

We use two independent atmospheric-retrieval codes to perform free-chemistry retrievals on the dayside thermal emission spectrum of WASP-18b: HYDRA[11,26,123,124], including the effects of thermal dissociation, and POSEIDON[125,126], which here assumes constant-with-depth abundances for all chemical species.

HYDRA consists of a parametric forward atmospheric model coupled to a Python implementation of the MultiNest[127] nested-sampling Bayesian parameter-estimation algorithm[128], PyMultiNest[103]. The inputs to the parametric model are the deep-atmosphere mixing ratios for each of the chemical species included, the temperature–pressure profile parameters (six free parameters) and a dilution parameter (area fraction) to account for 3D effects on the dayside[51]. Given the high dayside temperatures of WASP-18b, we consider high-temperature opacity sources and the effects of thermal dissociation, as described in ref. 11. We include opacity resulting from the molecular, atomic and ionic species with spectral features in the 0.8–2.8 μm range, which are expected in a high-temperature, $H_2$-rich atmosphere: collision-induced absorption owing to $H_2$–$H_2$ and $H_2$–He (ref. 129), $H_2O$ (ref. 112), CO (ref. 112), $CO_2$ (ref. 112), HCN (ref. 130), OH (ref. 112), TiO (ref. 116), VO (ref. 117), FeH (ref. 131), Na (ref. 110), K (ref. 110) and $H^-$ (refs. 107,108). The line-by-line absorption cross-sections for these species are calculated following the methods in ref. 132, using data from the references listed. The $H^-$ free-free and bound-free opacity is calculated using the methods in refs. 107,108, respectively. HYDRA includes the effects of thermal dissociation of $H_2O$, TiO, VO and $H^-$ as a function of pressure and temperature, following the method in ref. 9. In particular, the abundance profiles of these species are calculated following equations (1) and (2) in ref. 9, in which the deep abundance of each species ($A_0$) is a parameter in the retrieval and the $\alpha$, $\beta$ and $\gamma$ parameters are those given

in Table 1 of that same work. The abundance profiles of the remaining chemical species are assumed to be constant with depth.

HYDRA uses the parametric temperature–pressure profile in ref. 133, which has been used extensively in exoplanet atmosphere retrievals, including ultra-hot Jupiters such as WASP-18b (ref. 11). The temperature parametrization is able to capture thermally inverted, non-inverted and isothermal profiles, spanning the range of possible thermal structures for ultra-hot Jupiters. The HYDRA retrievals also include an area fraction parameter, $A_{HS}$, which multiplies the emergent emission spectrum by a constant factor to account for the dominant contribution of the hotspot[51]. Given the input chemical abundances, temperature–pressure profile parameters and area fraction, the model thermal emission spectrum is calculated at a resolving power of $R \approx 15,000$, convolved to the instrument resolution, binned to the data resolution and compared with the data to calculate the likelihood of the model instance. Detection significances are calculated for specific chemical species by comparing the Bayesian evidences of retrievals, which include/exclude the species in question[105]. These detection significances factor in the ability of the retrieval to fit the observations with a different temperature profile and/or other chemical species, when the species in question is not included. Because thermal emission spectra are very sensitive to the atmospheric temperature profile, changes in the temperature structure can, in some cases, slightly compensate for the absence of a particular chemical species, contributing to a lower detection significance.

We also use HYDRA to test the sensitivity of the free-chemistry retrievals to the limits of the log-normal prior distributions assumed for the chemical abundances. For the HYDRA retrieval including thermal-dissociation effects, we test two scenarios: wide, uninformative priors for all 11 species included (log mixing ratio ranging from −12 to −1) and slightly more restricted priors for the refractory species included (log mixing ratio ranging from −12 to −4 for TiO, VO and FeH, −12 to −2 for Na and K and −12 to −1 for the remaining species). The more restricted prior limits for the refractory species are motivated by the relatively lower abundances expected for these species compared with the volatile species, across a range of metallicities and C/O ratios[134–136].

We find that the atmospheric properties retrieved with HYDRA are consistent within $1\sigma$ for the two choices of prior limits. With the wide priors, the HYDRA retrieval infers a $H_2O$ log mixing ratio of $-3.09^{+1.28}_{-0.32}$, with double-peaked posterior probability distributions for the $H_2O$ and TiO abundances. Although the dominant posterior peaks correspond to approximately solar $H_2O$ and TiO abundances, the second, lower-likelihood peaks correspond to about 30 times and about $10^4$ times supersolar $H_2O$ and TiO abundances, respectively. Such an extreme TiO abundance warrants scepticism and may, for example, be a result of the well-known degeneracy between chemical abundances and the atmospheric temperature gradient (see also ref. 5). The retrieved $H_2O$ abundance in this case is consistent with the chemical-equilibrium retrievals and self-consistent 1D radiative–convective models described above, although with a larger uncertainty owing to the double-peaked posterior distribution. When the restricted priors are used, the low-likelihood, high-abundance peaks in the $H_2O$ and TiO posterior distributions are no longer present, and the retrieved $H_2O$ abundance is $-3.23^{+0.45}_{-0.29}$, in excellent agreement with the chemical-equilibrium retrievals and self-consistent 1D radiative–convective models. We note that the retrieved temperature–pressure profiles and abundances of CO, $CO_2$, HCN, OH, $H^-$ and FeH are unaffected by the choice of prior limits discussed above. The abundances of Na and K are unconstrained in both cases. Although the two choices of prior limits give consistent results, the expectation of chemical equilibrium in the atmosphere of WASP-18b, as well as the unlikelihood of an approximately $10^4$ times supersolar TiO abundance, motivate the use of the restricted priors on the refractory chemical abundances.

We note that, for either choice of prior, the retrieved deep-atmosphere abundance of VO is substantially higher than that inferred by the

chemical-equilibrium retrieval (Extended Data Fig. 5). This is because of a difference in the thermal-dissociation prescriptions; in the HYDRA retrieval, thermal dissociation results in a markedly depleted VO abundance at the photosphere, whereas in the chemical-equilibrium retrieval, thermal dissociation begins at lower pressures. Furthermore, the posterior distribution for the VO abundance peaks at highly super-solar values (about $10^4$ times solar). Such a high abundance is physically unlikely and may indicate the presence of further sources of optical opacity not included in the retrieval.

We also conduct free-chemistry retrievals, without the inclusion of thermal dissociation, using POSEIDON. POSEIDON is an atmospheric-retrieval code originally designed for the interpretation of exoplanet transmission spectra[125]. We have recently extended POSEIDON to include secondary-eclipse emission spectra modelling and retrieval capabilities. For an ultra-hot Jupiter such as WASP-18b, in which the dayside can be assumed clear, the emission forward model of POSEIDON calculates the emergent flux by means of a standard single-stream prescription without scattering

$$I_{\text{layer top}}(\mu,\lambda) = I_{\text{layer bot}}(\mu,\lambda)\,e^{-d\tau_{\text{layer}}(\lambda)/\mu} + (1-e^{-d\tau_{\text{layer}}(\lambda)/\mu})B(T_{\text{layer}},\lambda) \quad (8)$$

in which $I_{\text{layer bot}}$ and $I_{\text{layer top}}$ are, respectively, the upwards specific intensity incident on the lower layer boundary and the intensity leaving the upper layer boundary, $d\tau_{\text{layer}}$ is the differential vertical optical depth across the layer, $\mu = \cos\theta$ specifies the ray direction and $B(T_{\text{layer}},\lambda)$ is the blackbody spectral radiance at the layer temperature. Using the boundary condition $I_{\text{deep}}(\mu,\lambda) = B(T_{\text{layer}},\lambda)$, POSEIDON propagates equation (8) upwards to determine the emergent intensity at the top of the atmosphere. The emergent planetary flux is determined through

$$F_{\text{p,emergent}}(\lambda) = 2\pi\int_0^1 \mu I_{\text{emergent}}(\mu,\lambda)d\mu \approx 2\pi\sum_\mu \mu I_{\text{emergent}}(\mu,\lambda)W(\mu) \quad (9)$$

in which $W$ are the Gaussian quadrature weights corresponding to each $\mu$ (taken here as second-order quadrature over the interval $\mu \in [0,1]$). Finally, the planet–star flux ratio seen at Earth is given by

$$\left(\frac{F_p}{F_*}\right)_{\text{obs}}(\lambda) = \left(\frac{R_{\text{p,phot}}(\lambda)}{R_*}\right)^2 \frac{F_{\text{p,emergent}}(\lambda)}{F_*(\lambda)} \quad (10)$$

in which $R_{\text{p,phot}}$ is the effective photosphere radius[137,138] at wavelength $\lambda$ (evaluated at $\tau(\lambda) = 2/3$). Because the calculation of $R_{\text{p,phot}}$ requires a reference radius boundary condition to solve the equation of hydrostatic equilibrium, we prescribe $r(P = 10 \text{ mbar})$ as a free parameter.

For the WASP-18b POSEIDON retrieval analysis, we calculated emission spectra through opacity sampling and explored the parameter space using MultiNest through its Python wrapper PyMultiNest[103,127]. POSEIDON solves the radiative transfer on an intermediate-resolution wavelength grid (here, $R = 20,000$ from 0.8 to 3.0 μm), onto which high-spectral-resolution ($R \approx 10^6$), pre-computed cross-sections[139] are downsampled. For WASP-18b, we consider the following opacity sources: $H_2$–$H_2$ (ref. 140) and $H_2$–He (ref. 140) collision-induced absorption, $H_2O$ (ref. 111), CO (ref. 141), $CO_2$ (ref. 142), HCN (ref. 115), $H^-$ (ref. 108), OH (ref. 143), FeH (ref. 118), TiO (ref. 116), VO (ref. 117), Na (ref. 144) and K (ref. 144). We prescribed uniform-in-altitude mixing ratios, defined by a single free parameter for each of the chemical species included. The PyMultiNest retrievals with POSEIDON use 2,000 live points and the six-parameter temperature–pressure profile[133] outlined above in the description of HYDRA. POSEIDON accounts for the dominant contribution of the hotspot by prescribing the 10 millibar radius as a free parameter, which is subsequently converted into an equivalent $A_{\text{HS}}$ posterior by comparison with the white-light planet radius.

We note that both HYDRA and POSEIDON yield consistent retrieval results when thermal dissociation is not considered in the HYDRA

retrievals. However, the inclusion of thermal dissociation results in substantially different retrieved $H_2O$ abundances (see Extended Data Fig. 5) and temperature–pressure profiles, which are in agreement with the chemical-equilibrium retrievals and self-consistent 1D radiative–convective models described above.

## Disequilibrium chemistry model

To further justify the use of chemical-equilibrium models in our analysis of WASP-18b, we produce a grid of disequilibrium chemistry forward models to assess whether disequilibrium effects might strongly shape our observations. We begin by calculating the atmospheric temperature–pressure structure of WASP-18b under radiative–convective equilibrium with the HELIOS[145,146] radiative-transfer code. Next, we calculate altitude-dependent mixing ratios of chemical species under this temperature–pressure structure with the VULCAN[147] 1D chemical-kinetics code, using an N–C–H–O reaction network that includes ionization and recombination of Fe, Mg, Ca, Na, K, H and He. We use the current version of VULCAN (VULCAN2 (ref. 148)), which includes optional photochemistry and parametrizes the transport flux of chemical species with eddy diffusion, molecular diffusion, thermal diffusion and vertical advection. We revise this code to include the effect of photoionization. Finally, we generate emission spectra with the PLATON radiative-transfer code[149] at the resolution and wavelength range of NIRISS/SOSS. Our PLATON emission spectrum calculations use the code branch that allows varying chemical mixing ratios as a function of altitude[150]. We modify PLATON to calculate bound-free and free-free $H^-$ opacity as a function of altitude; this alteration is necessary to assess whether disequilibrium abundance $H^-$ opacity could mute spectral features more strongly than predicted by equilibrium chemistry. Furthermore, we modify PLATON to accept higher-temperature ($T > 3,000$ K) opacity files that we calculate with the HELIOS-K code[151,152].

For our set of models, we vary the eddy diffusion coefficient, $K_{zz}$, from $10^7$ cm$^{-2}$ s$^{-1}$ to $10^{13}$ cm$^{-2}$ s$^{-1}$, holding it constant at all altitudes for a given simulation. We perform this sweep over many orders of magnitude of $K_{zz}$ to understand the maximum effect that disequilibrium chemistry could have on the observed emission spectrum. Although $K_{zz}$ is a limited descriptor of vertical mixing and is expected to vary as a function of altitude (for example, ref. 153), we assume that our forward models bracket the expected vertical-mixing behaviour of this planet. This statement is further motivated by our GCMs, if we approximate $K_{zz} = vH$ for vertical wind velocity $v$ and atmospheric-scale height $H$ (refs. 154,155). The minimum dayside-average $K_{zz}$ for our kinematic MHD GCM (see 'GCMs' section) is about $10^8$ cm$^{-2}$ s$^{-1}$ and the maximum dayside-average $K_{zz}$ is about $10^9$ cm$^{-2}$ s$^{-1}$, well within our VULCAN grid range. Our model grid also toggles the inclusion of molecular diffusion and photochemistry. As input to VULCAN when photochemistry is included, we use a stellar spectrum that is appropriate for WASP-18 from ref. 156. The spectrum is constructed by joining synthetic spectra from ref. 157 and ultraviolet flux measurements of Piscium HD 222368 by the International Ultraviolet Explorer at 300 nm.

We find that our inclusion of disequilibrium chemistry effects—photochemistry, molecular diffusion, thermal diffusion and eddy diffusion—produces spectra that are not strongly discrepant from spectra computed assuming chemical equilibrium. Indeed, all discrepancies between spectra produced under chemical equilibrium and spectra produced under chemical disequilibrium spectra are less than 10 ppm. This agreement is expected, as chemistry at the pressure levels examined by low-resolution emission spectra are not predicted to be strongly modified by photochemistry or mixing (for example, ref. 148). Furthermore, the high temperature of WASP-18b implies that the chemical timescales in the atmosphere are short, allowing chemical reactions to occur more quickly than the relevant disequilibrium timescales.

Overall, our grid of chemical disequilibrium forward models indicates that disequilibrium chemistry effects considered here do not

strongly affect the emission spectrum of WASP-18b in the NIRISS/SOSS waveband.

## GCMs

A suite of GCMs is compared with the retrieved dayside spectrum and dayside temperature map.

We use the SPARC/MITgcm[53] to model the 3D atmospheric structure of WASP-18b. The model solves the primitive equations in spherical geometry using the MITgcm[158] and the radiative-transfer equations using a current 1D radiative-transfer model[159]. We use the correlated-$k$ framework to generate opacities based on the line-by-line opacities[160]. Our model assumes a solar composition for the elemental abundances[161] and chemical equilibrium gas-phase composition[162]. Our model naturally takes into account the effect of thermal dissociation[9]. We used a time step of 25 s and ran the simulations for about 300 Earth days, averaging all quantities over the last 100 days.

We include further sources of drag through a Rayleigh drag parametrization with a single constant timescale per model that determines the efficiency with which the flow is damped. The drag is constant over the whole planetary atmosphere. We vary this timescale between models from $\tau_{\mathrm{drag}} = 10^3$ to $10^6$ s (efficient drag) and a no-drag model with $\tau_{\mathrm{drag}} = \infty$. Our range of drag strengths cover the transition from a drag-free, wind-circulation case to a drag-dominated circulation. The specific WASP-18b simulations that we use are described in more detail in ref. 16.

The second model we use is the kinematic MHD GCM (described in detail in ref. 47) with a revised picket-fence radiative-transfer scheme[163]. Owing to the high gravity of this planet, we chose to model the planet from 100 bar to $10^{-4}$ bar over 65 layers at a horizontal resolution of T31 (corresponding to roughly 3° resolution at the equator). We use the kinematic MHD prescription described in ref. 44, which has been used in models of hot Jupiters HD 209458b and HD 189733b (ref. 164), as well as the ultra-hot Jupiter WASP-76b (refs. 47,52). This drag prescription assumes a global dipole magnetic field, generated by an interior dynamo. Because of this geometry, our drag timescale is applied as a Rayleigh drag term solely to the east–west momentum equation (influencing flow perpendicular to magnetic field lines) and is calculated as

$$\tau_{\mathrm{mag}}(B, \rho, T, \phi) = \frac{4\pi\rho \, \eta(\rho, T)}{B^2 \, |\sin(\phi)|}, \tag{11}$$

in which $B$ is the chosen global magnetic field strength, $\phi$ is the latitude, $\rho$ is the density and $\eta$ is magnetic resistivity. This timescale is calculated locally and often, allowing the timescale to vary by more than ten orders of magnitude throughout the model and respond to changes in atmospheric temperatures. A minimum timescale cutoff (roughly $10^3$ s) is applied in locations in which $\tau_{\mathrm{mag}}$ would be less than 1/20th of the planet's orbit, for numerical stability. We chose to model a range of magnetic field strengths (0, 5, 10 and 20 G), as its true value is not known.

## Data availability

The data used in this work are publicly available in the Mikulski Archive for Space Telescopes (MAST) (https://archive.stsci.edu/). The data that were used to create all of the figures in this manuscript are freely available on Zenodo (https://doi.org/10.5281/zenodo.7907569). All further data are available on request. Source data are provided with this paper.

## Code availability

The open-source pipelines that were used throughout this work are as follows: NIRISS/SOSS data reduction: nirHiss (https://github.com/afeinstein20/nirhiss); supreme-SPOON (https://github.com/radicamc/supreme-spoon); transitspectroscopy (https://github.com/nespinoza/transitspectroscopy). Light-curve fitting: batman (https://github.com/lkreidberg/batman); emcee (https://emcee.readthedocs.io/en/stable/). Atmospheric retrievals: CHIMERA (https://github.com/mrline/CHIMERA); POSEIDON (https://github.com/MartianColonist/POSEIDON); MultiNest (https://github.com/JohannesBuchner/MultiNest); PyMultiNest (https://github.com/JohannesBuchner/PyMultiNest). Eclipse mapping: ThERESA (https://github.com/rychallener/ThERESA); Eigenspectra (https://github.com/multidworlds/eigenspectra). Atmospheric modelling: HELIOS (https://github.com/exoclime/HELIOS); HELIOS-K (https://github.com/exoclime/HELIOS-K); PLATON (https://github.com/ideasrule/platon); VULCAN (https://github.com/exoclime/VULCAN).

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

**Acknowledgements** This work is based on observations made with the NASA/ESA/CSA JWST. The data were obtained from the Mikulski Archive for Space Telescopes (MAST) at the Space Telescope Science Institute, which is operated by the Association of Universities for Research in Astronomy, Inc., under NASA contract NAS 5-03127. These observations are associated with program JWST-ERS-01366. Support for this program was provided by NASA through a grant from the Space Telescope Science Institute. The results reported herein benefited during the design phase from collaborations and/or information exchange within NASA's Nexus for Exoplanet System Science (NExSS) research coordination network sponsored by NASA's Science Mission Directorate. L.-P.C. acknowledges funding by the Technologies for Exo-Planetary Science (TEPS) Natural Sciences and Engineering Research Council of Canada (NSERC) CREATE Trainee Program. M.M., R.J.M. and L.W. acknowledge support provided by NASA through the NASA Hubble Fellowship Program. A.D.F. acknowledges support from the National Science Foundation through the Graduate Research Fellowship Program. I.W. acknowledges support provided by NASA through the NASA Postdoctoral Program. B.V.R. acknowledges support from the 51 Pegasi b Fellowship funded by the Heising-Simons Foundation.

**Author contributions** All authors played a notable role in one or more of the following: development of the original proposal, management of the project, definition of the target list and observation plan, analysis of the data, theoretical modelling and preparation of this manuscript. Some specific contributions are listed as follows. N.M.B., J.L.B. and K.B.S. provided overall program leadership and management. L.-P.C. and B.B. led the efforts for this manuscript. D.K.S., E.M.-R.K., H.A.K., H.R.W., I.J.M.C., J.L.B., J.-M.D., K.B.S., L.K., M.L.-M., M.R.L., N.M.B., N.C., V.P. and Z.K.B.-T. made substantial contributions to the design of the program. K.B.S. generated the observing plan, with input from the team. N.E. provided instrument expertise. B.B., E.M.-R.K., H.R.W., I.J.M.C., J.L.B., L.K., M.L.-M., M.R.L., N.M.B. and Z.K.B.-T. led or co-led working groups and/or contributed to substantial strategic planning efforts, such as the design and implementation of the pre-launch Data Challenges. A.L.C., D.K.S., N.E., N.P.G. and V.P. generated simulated data for pre-launch testing of methods. B.B., E.R., J.L.B., L.-P.C., T.D.K. and V.P. contributed notably to the writing of this manuscript, along with contributions in the Methods from A.A.A.P., A.B.S., A.D.F., H.B., I.W., L.A.D.S., L.S.W., M.M., M.R., R.C., R.J.M. and X.T. A.D.F., L.A.D.S., L.-P.C. and M.R. contributed to the development of data-analysis pipelines and/or provided the data-analysis products used in this analysis, that is, reduced the data, modelled the light curves and/or produced the planetary spectrum. I.W. performed the revised TESS analysis. A.A.A.P., L.-P.C., L.S.W. and R.J.M. performed atmospheric retrievals on the planetary spectrum. M.H., M.M., N.T.L. and R.C. performed the secondary-eclipse-mapping analysis. E.K.H.L., H.B., L.-P.C., R.C., V.P. and X.T. provided GCM and/or post-processed GCM results used in this analysis. A.B.S. generated the disequilibrium chemistry models. H.S. and L.-P.C. produced the absolute flux-calibrated stellar spectrum and performed the comparison with stellar models. L.-P.C., L.S.W. and R.C. generated the figures for this manuscript. B.V.R., H.R.W., I.J.M.C., J.H., L.W., M.C.N. and X.Z. provided substantial feedback to the manuscript and/or coordinated comments from all other authors. A.S., C.C., C.P., E.-M.A., E.P., J.B., J.J.F., J.M.G., J.L., J.D.L., K.A., K.H., K.M., K.O., L.D., L.M., M.K.A., M.L., N.I., N.K.N., O.V., P.E.C., P.-O.L., P.-A.R., P.J.W., Q.C., S.L.C., S.K., S.R., S.-M.T., T.J.B., T.H. and T.M.-E. provided scientific and technical input to the manuscript.

**Competing interests** The authors declare no competing interests.

## Additional information

**Correspondence and requests for materials** should be addressed to Louis-Philippe Coulombe.

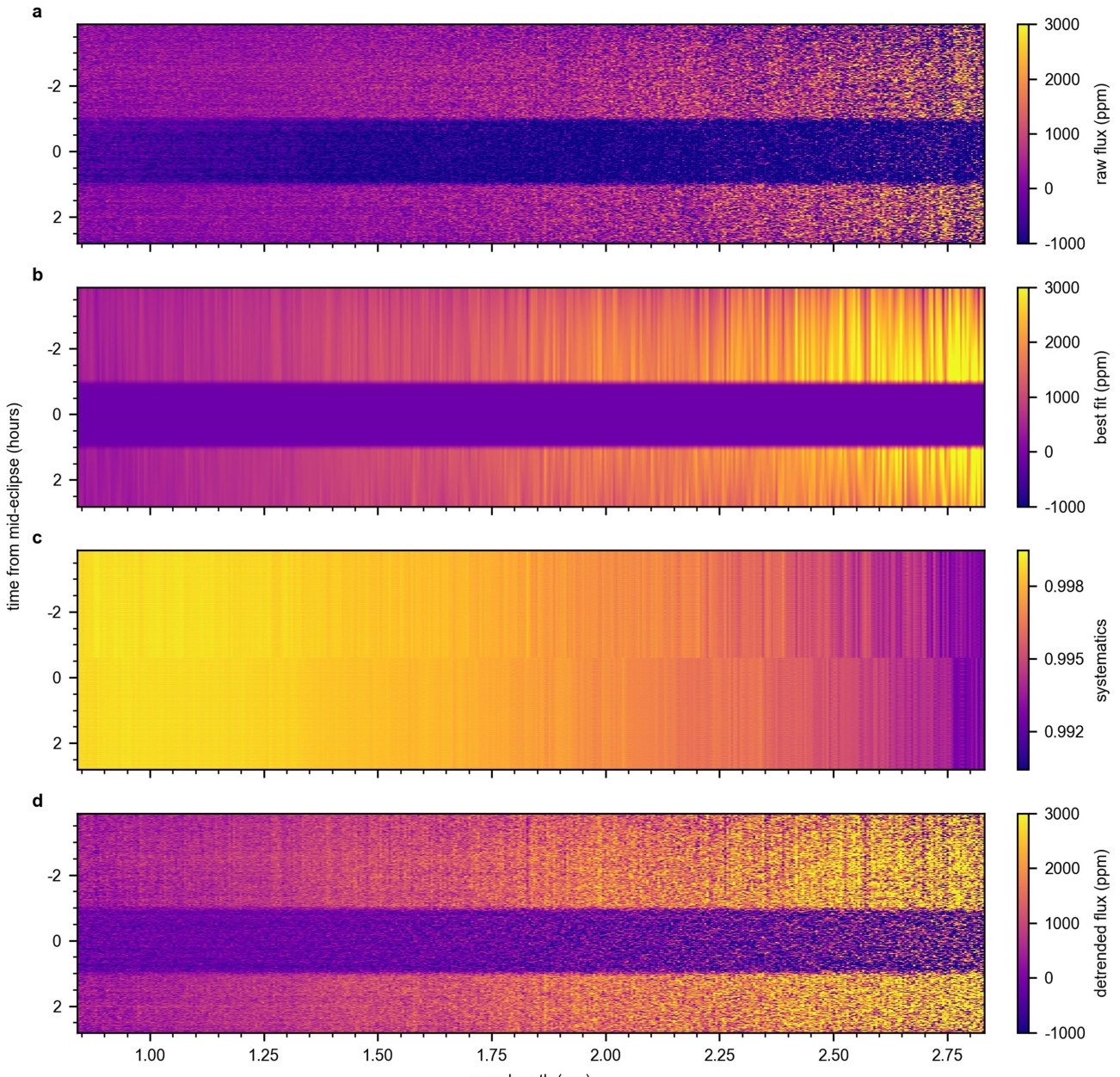

**Extended Data Fig. 1 | Spectrophotometric secondary-eclipse light curves of WASP-18b. a**, Raw light curves for all 408 spectrophotometric bins. **b**, Best-fit planetary flux measured from the light-curve fits. **c**, Systematics subtracted from **a**, consisting of a linear trend and the detrending against the tilt event and the trace morphology changes. The jump around 0.7 h before mid-eclipse comes from the fit of the flux offset caused by the tilt event. **d**, Raw light curves after subtraction of the best-fit systematics model. Some of the detrended light curves show sudden flux variations between wavelength bins outside of eclipse caused by correlations between the astrophysical and systematics models. Those correlations are, however, considered when computing the spectrum, as the $F_p/F_s$ values are marginalized over the range of systematics model that fit the light curves.

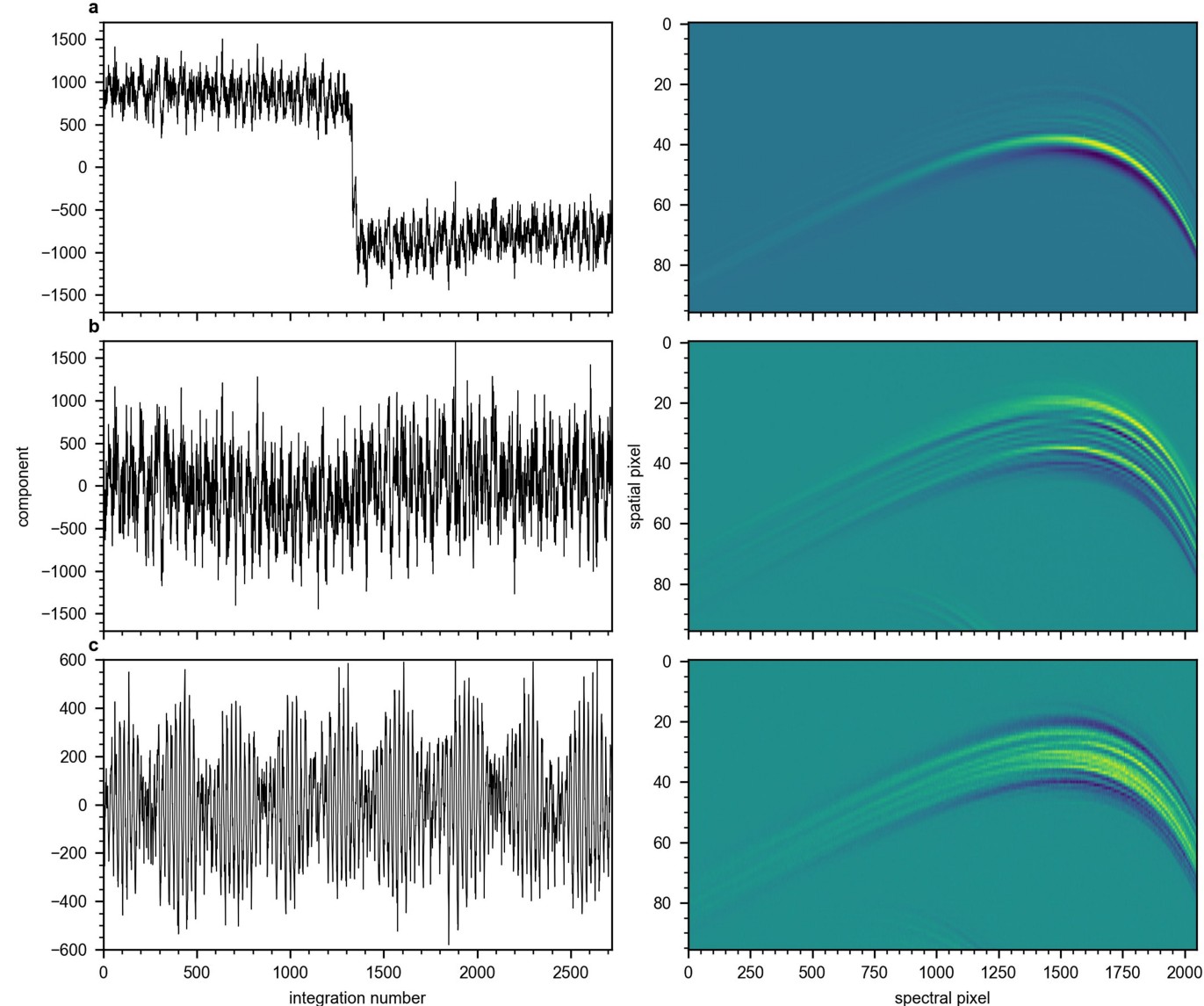

**Extended Data Fig. 2 | Morphological changes of the spectral trace on the NIRISS detector as identified through PCA on the time series of the detector images. a**, First principal component with its eigenvalues (left) and its corresponding eigenimage (right). The tilt event occurring near the 1,336th integration can clearly be identified as the largest source of variance to the detector images. It results in a subtle change to the trace profile in the cross-dispersion direction, predominantly visible near its lower edge of the trace. **b**, Second principal component with its eigenvalues (left) and its corresponding eigenimage (right). The second principal component represents subtle changes in the y position of the trace throughout the time series, with the two edges of the trace trading flux. **c**, Third principal component with its eigenvalues (left) and its corresponding eigenimage (right). The third component represents changes in the FWHM of the trace and shows a clear beat pattern in time. The eigenimage for this component shows a trade of flux between the centre and the edges of the trace throughout the time series.

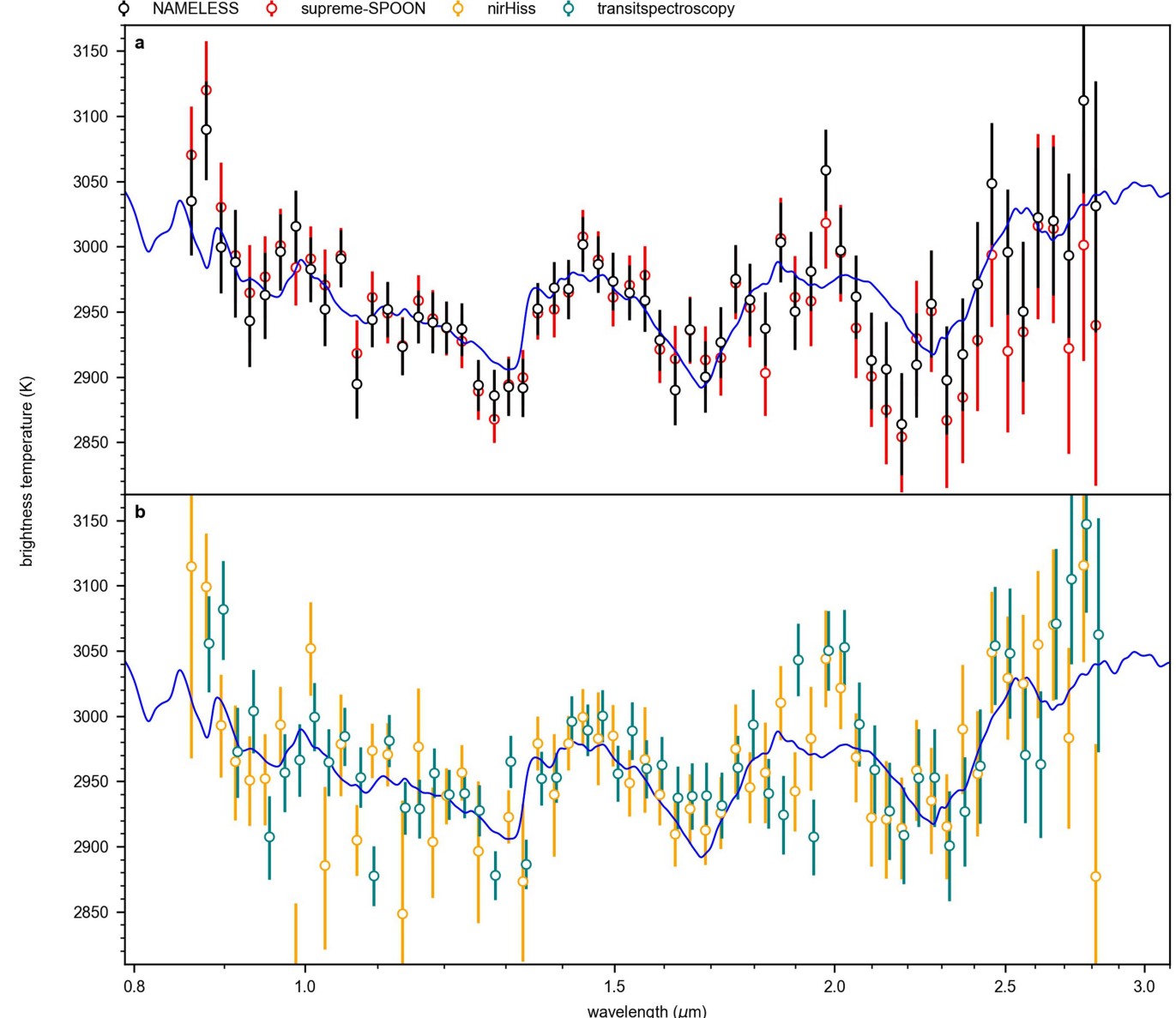

**Extended Data Fig. 3 | Spectra from the four individual reductions.** Comparison of the brightness temperature spectra obtained by fitting with ExoTEP the four separate reductions and binned at a resolving power of $R = 50$. All data are plotted with their $1\sigma$ error bars. We overplot the best-fit SCARLET model (blue line) to the reductions for further comparison. All reductions are consistent within less than one standard deviation on average when compared at full resolution (408 bins).

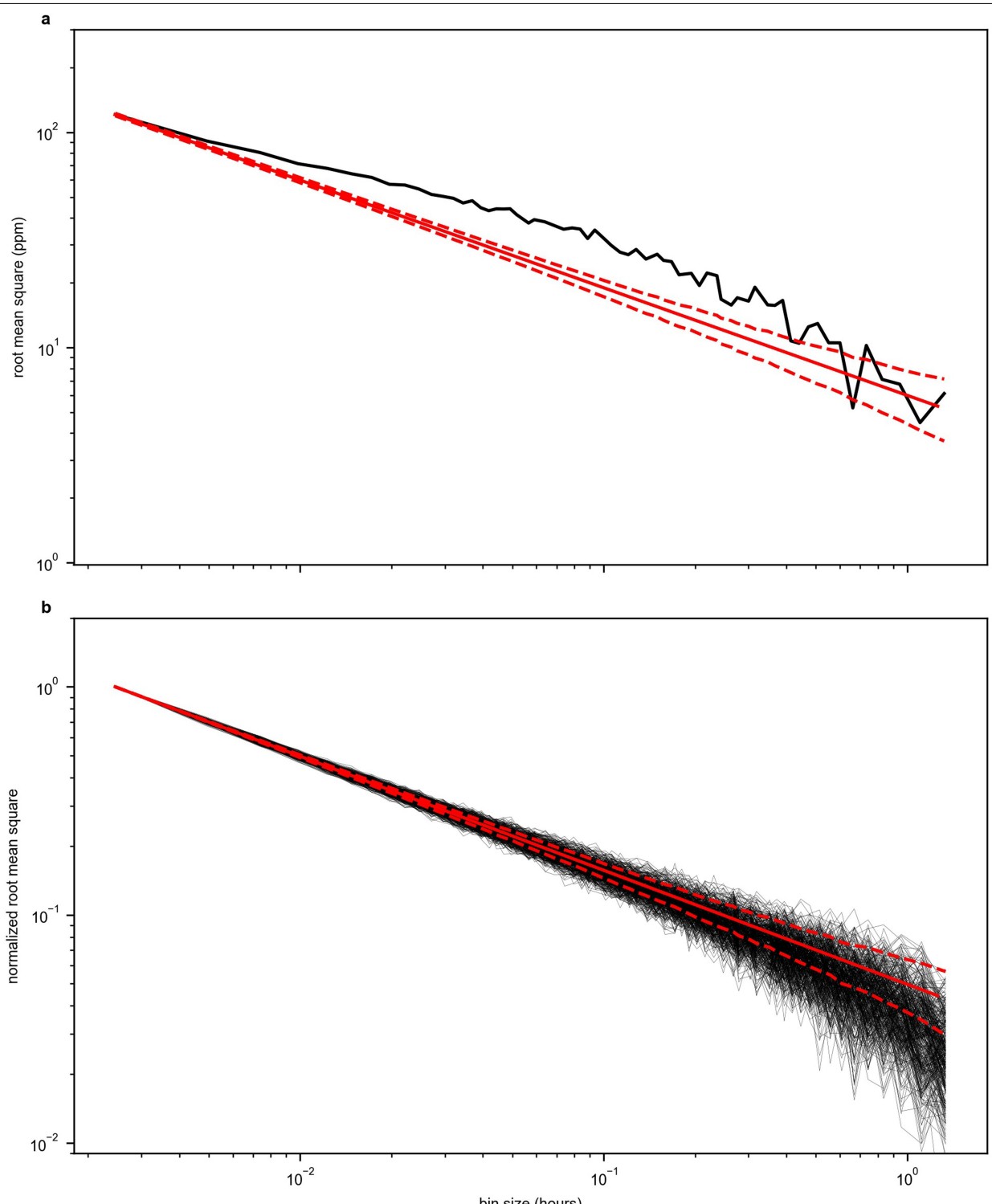

**Extended Data Fig. 4 | Light-curve residuals binned in time. a**, Absolute root mean square (RMS) of the residuals as a function of bin size (black line) for the white-light curve. The RMS values are plotted against the Poisson noise limit (red line), which decreases as the square root of the number of integrations contained in a single bin. We also show the theoretical $1\sigma$ error envelope of the Poisson noise. The residuals bin down to about 5 ppm for bins of 1 h and show no evidence of a noise floor, similar to what was observed from commissioning data[54]. The broadband residuals do not perfectly follow the Poisson noise, which is indicative of remaining time correlations. **b**, Normalized RMS of the 408 spectrophotometric light curves considered in the analysis. We observe that the residuals follow the Poisson noise limit from bin sizes of a single integration up to bins of approximately 1 h, indicating that there are no time correlations in the residuals. We observe a slight decrease of the normalized RMS below the Poisson noise at larger bin sizes, similar to what was observed in the NIRCam and NIRSpec/G395H observations of WASP-39b (refs. 21,165).

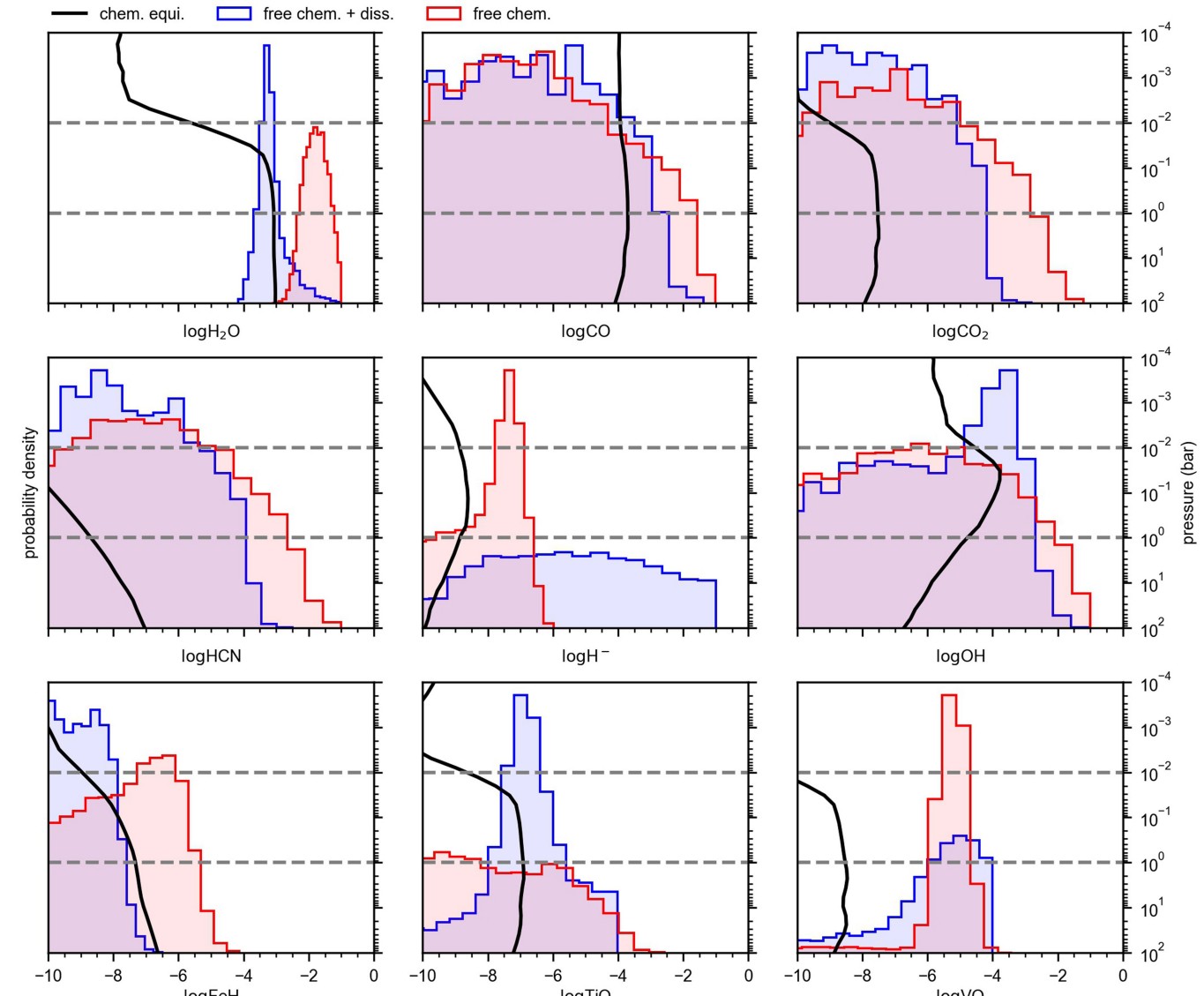

**Extended Data Fig. 5 | Abundance constraints from the free-chemistry retrievals.** Probability posteriors of the deep abundance of various species considered for the free chemistry and temperature with (blue, HyDRA) and without (red, POSEIDON) thermal dissociation. We also show the median retrieved VMR profiles from the chemical equilibrium with free temperature–pressure profile retrieval (black line, SCARLET). The pressure range investigated by the observations (about 0.01–1 bar; see Fig. 3) is indicated by the dashed grey lines. The only species independently detected is H$_2$O, which is found to be consistent with the retrieved chemical-equilibrium abundance when considering the effect of thermal dissociation. All other species considered are found to be unconstrained, although consistent with chemical-equilibrium predictions. The photosphere as predicted by our radiative–convective model is around 50 millibar, but the retrievals infer the deep molecular abundances.

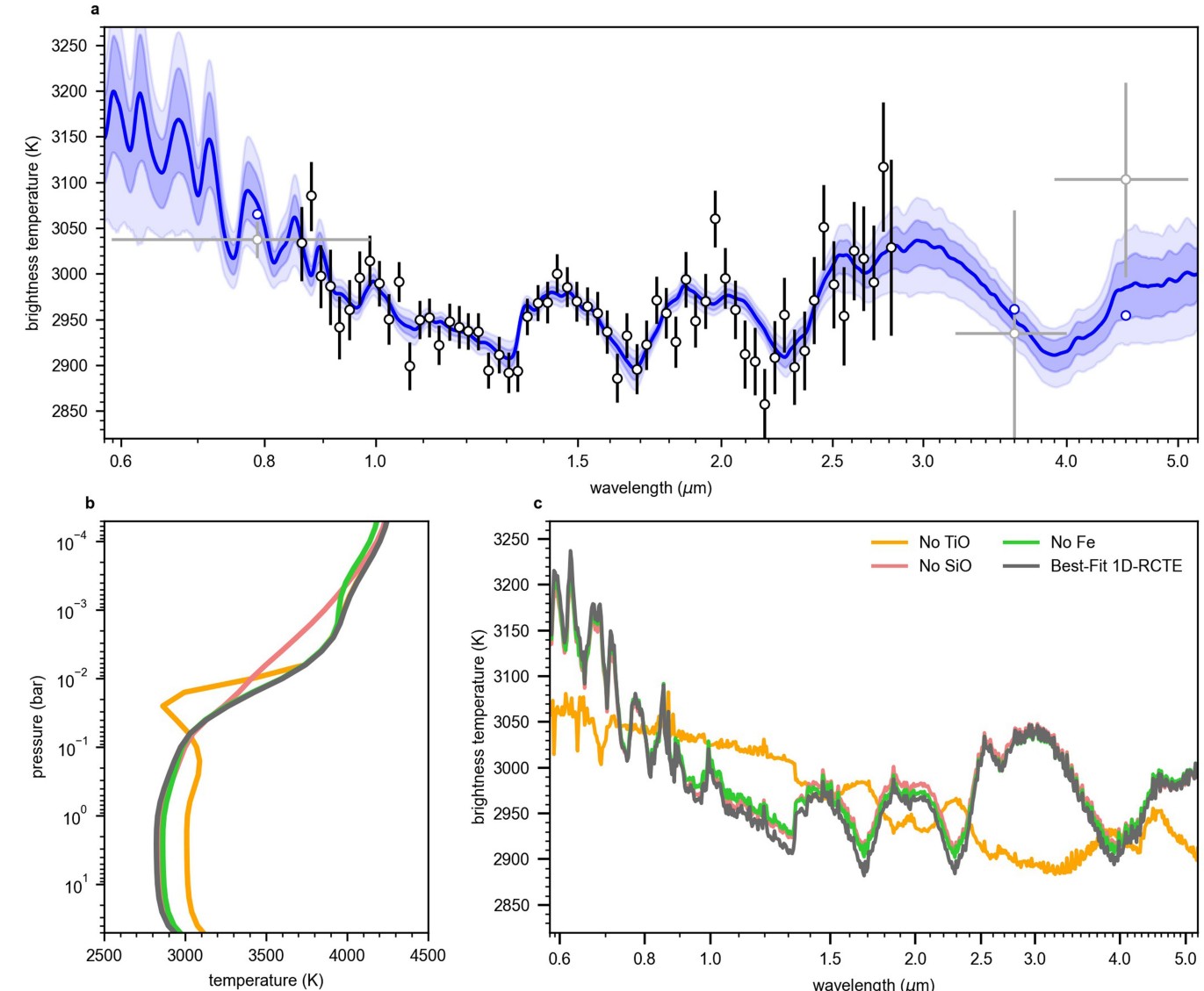

**Extended Data Fig. 6 | WASP-18b's brightness temperature spectrum fit and source of the thermal inversion. a**, The dark blue line indicates the chemical-equilibrium median fit to the NIRISS data with its $1\sigma$ error bars (black points), with shaded blue regions showing the $1\sigma$ and $2\sigma$ credible intervals in the retrieved spectrum (medium and light blue, respectively). The spectra are extrapolated to the TESS (visible wavelengths) and the Spitzer/IRAC measurements (3.6 and 4.5 μm) observations (grey points) considering the same atmospheric parameters. **b**, Best-fit radiative–convective model temperature–pressure profile together with radiative–convective solutions in which specific species known to create a thermal inversion are removed from the atmosphere. Absorption by atomic iron contributes to the thermal inversion at pressures lower than 1.0 millibar, whereas TiO is responsible for the thermal inversion seen between 0.1 and 0.01 bar. SiO contributes at pressures lower than 0.1 bar. **c**, Best-fit radiative–convective brightness temperature spectrum (excluding area fraction) and resulting spectra when removing specific species. As shown by the change from emission to absorption features in the spectra when TiO is removed, the TiO-induced thermal inversion is that examined by our observations.

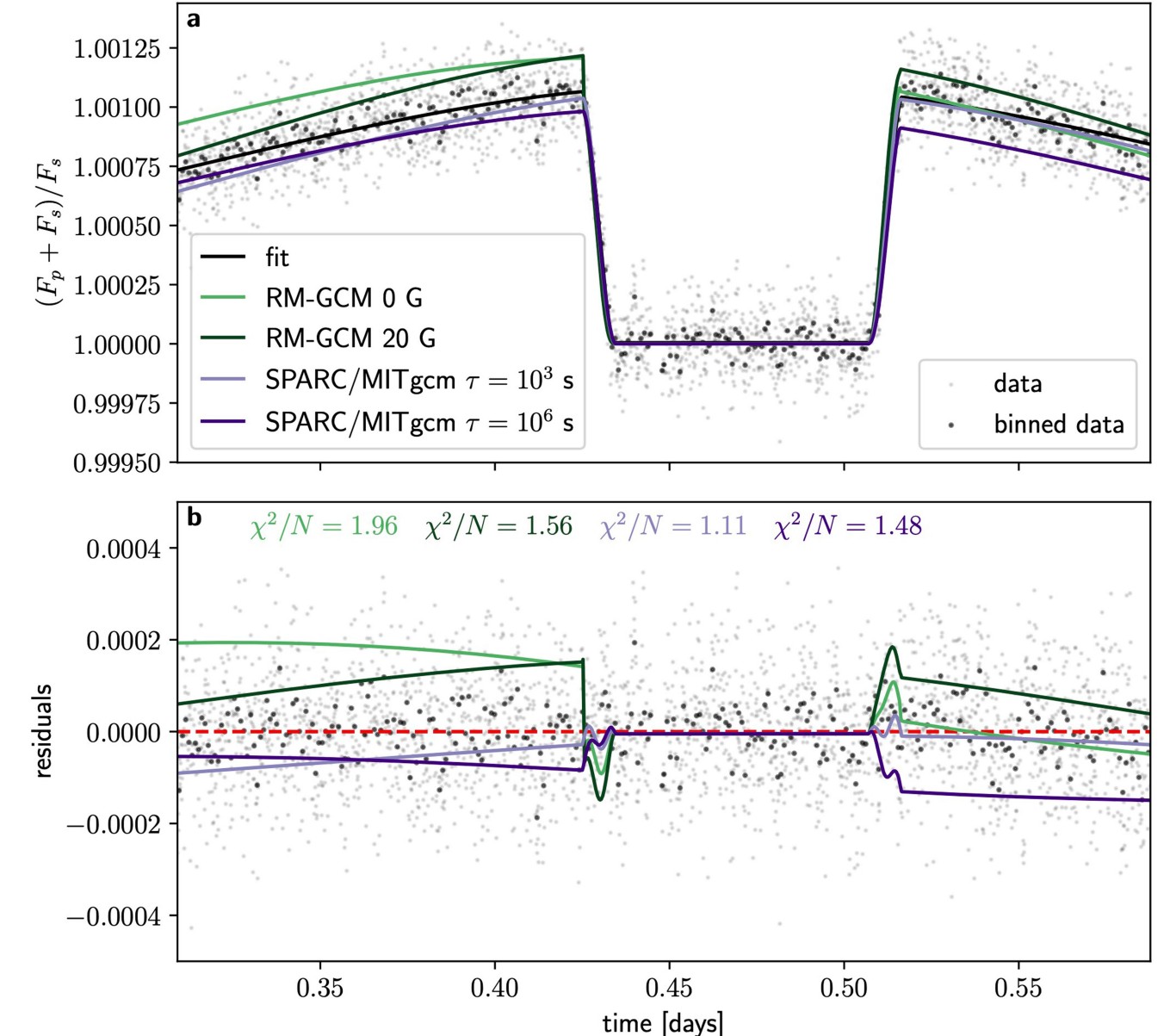

**Extended Data Fig. 7 | Light-curve predictions from GCMs. a**, GCM light curves compared against the data and the best-fitting eclipse-mapping model. **b**, Data and the GCM light curves with the eclipse-mapping model subtracted. The GCMs with strong atmospheric drag (RM-GCM $B = 20$ G and SPARC/MITgcm $\tau_{\mathrm{drag}} = 10^3$ s) match the data better than their counterparts that have little to no drag.

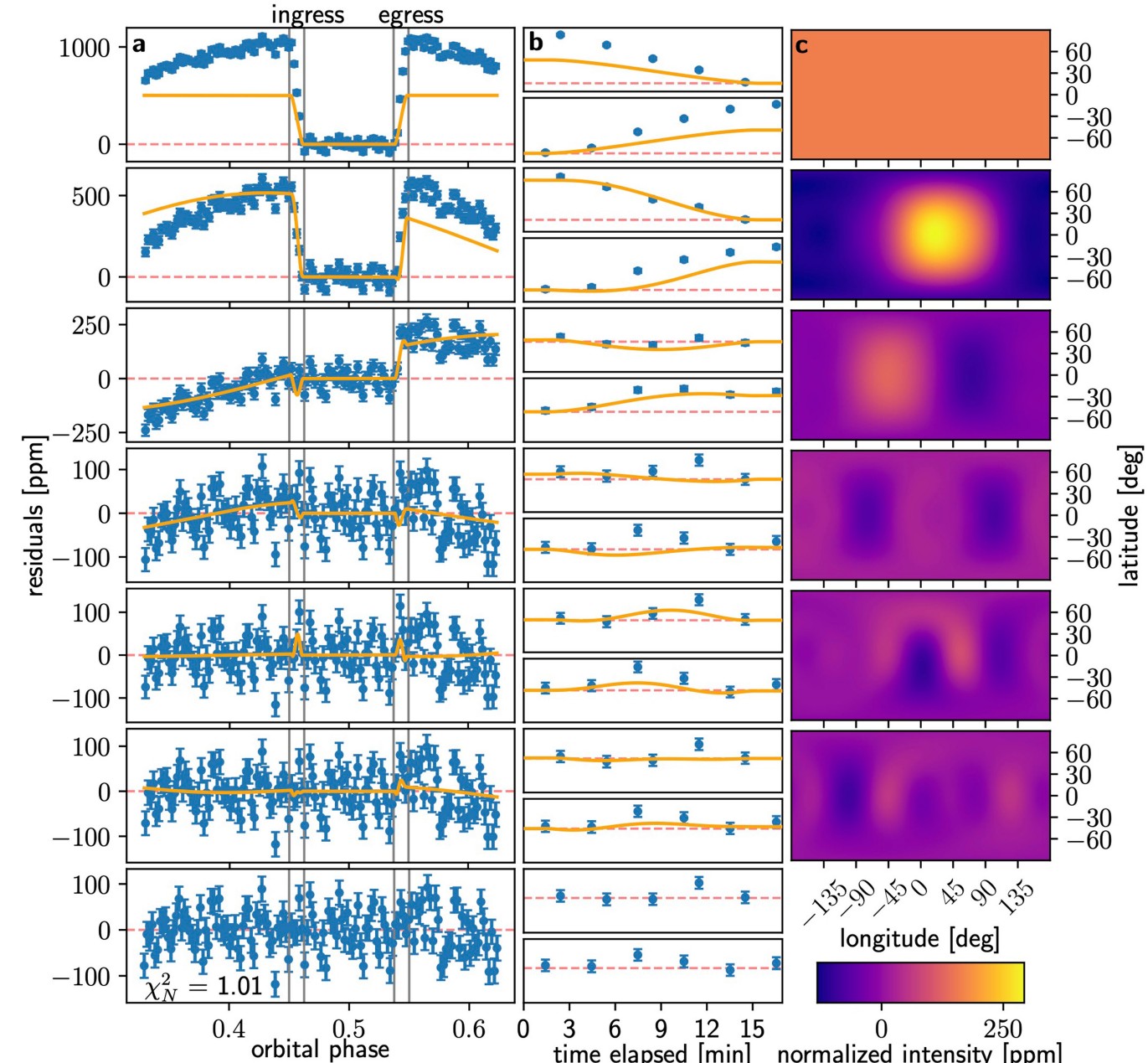

**Extended Data Fig. 8 | Components of the eclipse-mapping fit. a**, This column shows the light-curve components of the eclipse-mapping fit for $l_{max} = 5$, $N = 5$, overplotted on the data, which have been binned by a factor of 20 for clarity. From top to bottom, each light-curve component is subtracted from the data to illustrate the features that are fit by each component, such that the top row is the full white-light curve and the bottom row is the model residuals. The white-light curve points are plotted with their $1\sigma$ error bars. Note that all components are fit simultaneously. **b**, The same as column **a**, zoomed in to the ingress and egress of the eclipse to highlight the fine features fit by each component. **c**, The eigenmaps associated with the corresponding components in columns **a** and **b** that, when integrated, generate those light curves. Each map has been scaled by its best-fitting weight, such that a sum of this column would produce the best-fitting map.

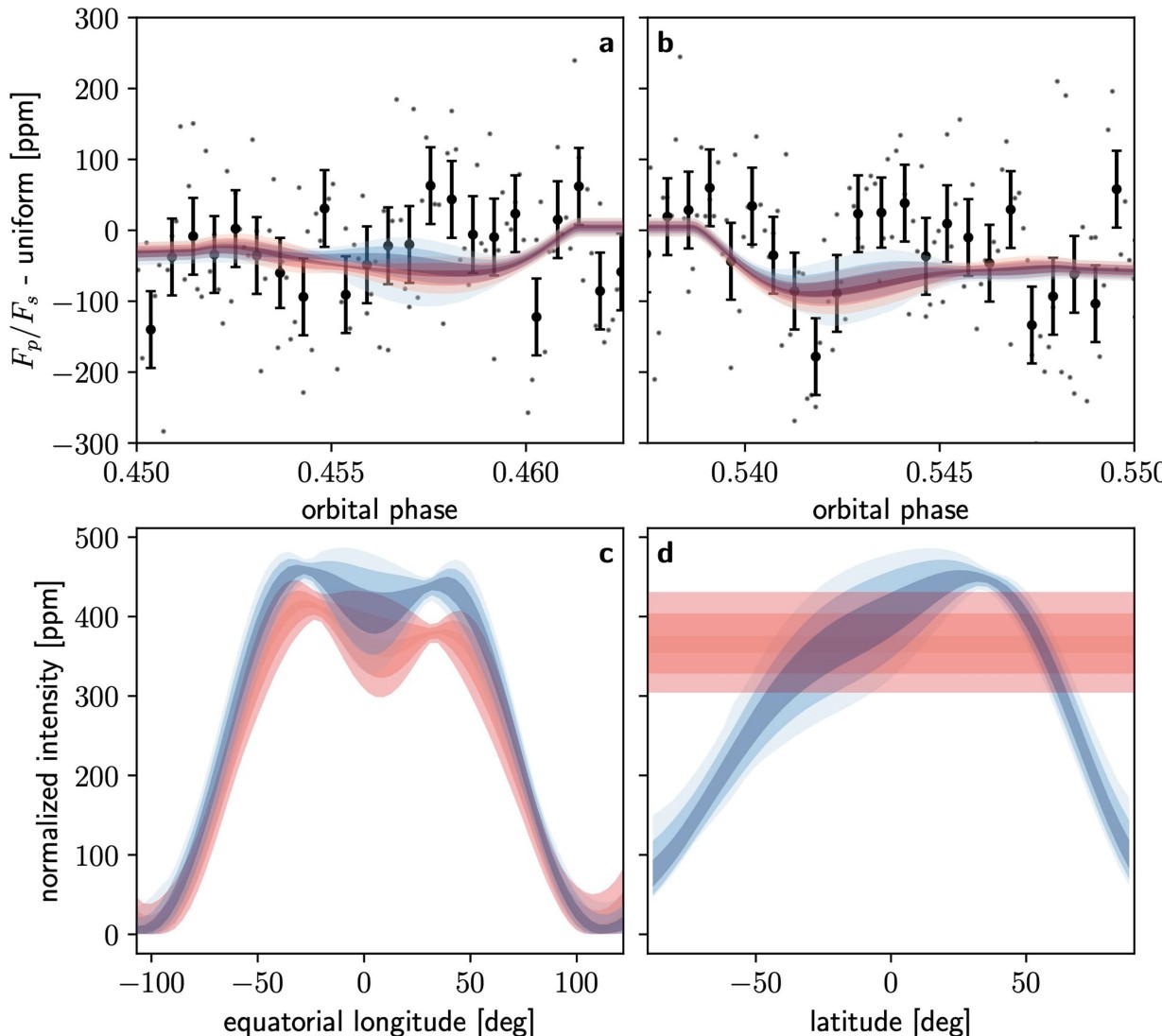

**Extended Data Fig. 9 | Latitudinal structure in the eclipse map. a**, Ingress of the eclipse, with two models overplotted and a 1,096 ppm (white-light planet flux at mid-eclipse) uniform planet model subtracted to highlight deviations. The data (small dots) have been binned by a factor of five (dots with 1σ error bars) for clarity. The blue model is the eclipse map for $l_{max} = 5$, $N = 5$ presented in the text. The red model uses a constant-with-latitude Fourier series as the basis set, rather than spherical harmonics, to investigate constraints on latitudinal aspects of the map. Shaded regions denote 1σ, 2σ and 3σ quantiles. **b**, Same as **a** but for the eclipse egress. **c**, Planetary flux along the equator for the same two models. Note that, regardless of basis functions, we retrieve the same longitudinal structure, giving us confidence in the longitudinal brightness distribution. **d**, Same as **c** but along the substellar meridian. Both models fit the data well but find different latitudinal structure, indicating that we are unable to constrain latitudinal variation.

**Extended Data Table 1 | TESS and JWST NIRISS/SOSS fit results**

| Parameter | TESS | JWST NIRISS/SOSS |
|---|---|---|
| $R_p/R_*$ | $0.09783 \pm 0.00028$ | $0.09783$ |
| $T_0$ [BJD$_{\rm TDB}$] | $2458747.985032 \pm 0.000027$ | – |
| $T_{sec}$ [BJD$_{\rm TDB}$] | – | $2459802.881867 \pm 0.000092$ |
| $P$ [days] | $0.941452382 \pm 0.000000069$ | $0.941452382$ |
| $b$ | $0.360 \pm 0.026$ | $0.340 \pm 0.018$ |
| $a/R_*$ | $3.496 \pm 0.029$ | $3.483 \pm 0.021$ |
| $u_1$ | $0.290 \pm 0.032$ | – |
| $u_2$ | $0.169 \pm 0.061$ | – |
| $\bar{f}_p$ [ppm] | $180 \pm 13$ | – |
| $F_{\rm atm}$ [ppm] | $177.5 \pm 5.7$ | – |
| $D$ [ppm] | $21.8 \pm 5.2$ | $21.8$ |
| $E$ [ppm] | $172.2 \pm 5.6$ | $172.2$ |
| Eclipse depth [ppm] | $357 \pm 14$ | $1096 \pm 38$ |
| Nightside flux [ppm] | $2 \pm 14$ | – |
| $i$ [deg] | $84.09 \pm 0.47$ | $84.39 \pm 0.30$ |

The median and 1σ uncertainties of the astrophysical parameters from the TESS (left column) and JWST NIRISS/SOSS (right column) analyses of WASP-18. The transit observations in the TESS phase curve are fitted considering the $u_1$ and $u_2$ quadratic limb-darkening coefficients. In our TESS phase-curve parametrization, the secondary-eclipse depth and nightside flux are derived parameters calculated from the average relative planetary flux $\bar{f}_p$ and the phase-curve semiamplitude $F_{\rm atm}$ of the planet. Likewise, the orbital inclination $i$ is derived from the scaled semi-major axis $a/R_*$ and the impact parameter $b$. The parameters fixed to the values from the TESS analysis for the NIRISS/SOSS light-curve fitting are shown without uncertainties in the table. The retrieved parameters are the time of secondary eclipse $T_{sec}$, the impact parameter $b$ and the semimajor axis $a/R_*$, the last two being fitted considering normal priors from the TESS constraints. The NIRISS/SOSS eclipse depth is derived from the samples of the astrophysical model (equation (4)).