## [Peer Review File · Nature]

Manuscript Title: A broadband thermal emission spectrum of the ultra-hot Jupiter WASP-18b

Editorial Notes:

Reviewer Reports on the Initial Version:

Referee expertise:

Referee #1: exoplanet observations

Referee #2: exoplanet observations

Referees' comments:

Referee #1 (Remarks to the Author):

This paper describes important and well executed results that will be of wide interest. The data and the analysis are of generally high quality, and I have no show-stopping criticisms. But this paper has an impressive amount of substance, and I think serious consideration should be given to dividing it into two papers - one paper dealing with the emergent day side spectrum and composition, and a second paper describing the map and implications for atmospheric drag. Both of those papers would (in my opinion) be sufficiently important to warrant publication in Nature. And two papers would be much more focused, readable, and citeable, than one paper.

Given the ample substance in this paper, I have quite a few issues that the authors should consider before the final version(s) is/are accepted:

lines 139, 410, and 468. I question whether fitting for the shape and duration of the eclipse is justified, as opposed to freezing those parameters based on TESS observations of the transit. The transit being a much stronger signal, and the TESS data being of high quality over many transits, it is arguably preferable to simply freeze the orbital parameters at their TESS values instead of fitting for them using the eclipse (even with narrow priors). On the other hand, the JWST data have exquisite precision, so maybe the authors are OK in this case, but that's not obvious. Moreover, the orbital parameters in Extended Table 2, and those quoted in line 468 aren't identical, albeit they are consistent within the errors. But it's the same planet in the same orbit, so why not constrain the exact same orbital parameters in all cases? I suggest that the authors should re-consider the fitting methods, and arguably the best and most consistent procedure would be to fit to the TESS transit data and the JWST data constrained with the exact same orbital parameters - including the white light curve, the spectral eclipses, and the eclipse map. Or at a minimum, clarify the process by summarizing how all of those fitted parameters relate to the TESS data that seem to be the primary source of the orbital parameters.

A major component that is missing from this paper is the timing, duration and depth of the secondary eclipse as observed by JWST. There's a Table for TESS results, but oddly there's no Table describing the eclipse parameters from JWST. Please list the JWST secondary eclipse time and duration in a prominent Table (and including the white-light depth and other parameters as appropriate).

When discussing the metallicity of the planet (your results are highly interesting), it would be appropriate to consider what you could say about possible limits on accretion during migration, quoting models as might be appropriate. This planet has evidently migrated a long distance, and it seems surprising that it didn't eat many comets and asteroids along the way. Can the authors usefully comment on that issue?

Line 228, in the brightness-temperature mapping, the authors assume that the star emits as a blackbody. That's a poor assumption, and that portion of the analysis should be re-done using model atmospheres. Interpolating in a model atmosphere grid for the star is a much better approximation than a blackbody star. There's no excuse for using a blackbody when high quality model atmospheres are available.

In the paragraph beginning on line 596, the discussion of the orbital eccentricity is confusing. The timing of the eclipse is more sensitive to the eccentricity than the RV observations are. Already there were strong limits placed on the eccentricity ($e \cdot \cos(\omega)$) by Nymeyer et al. and Maxted et al. And indeed the timing and duration of this JWST eclipse combined with photometry from TESS should provide very stringent limits on the eccentricity. Instead of quoting a possible eccentricity from RV measurements, I don't understand why you don't derive and use strong limits on the eccentricity from the TESS, Spitzer, and your own JWST photometry. Indeed, line 414 already said that you have a strong preference for zero eccentricity.

line 542, "it is possible to recover a fit which contains regions of negative planet emission. This is physically impossible, so we impose a positivity constraint on the total flux map." I am concerned that a positivity constraint will bias the eclipse maps, due to Lucy-Sweeny bias. Can the authors justify quantitatively that the results will not be biased?

The eclipse map (Figure 4) is done in white-light, but that bandpass has an effective wavelength. I suggest quoting the effective wavelength because JWST maps will doubtless be made in other bands. (the effective wavelength is derived from integrating the emergent flux weighted by the instrument sensitivity.)

Minor comments:

First paragraph, "whereas the SUBSTRIP256 subarray also provides the shorter-wavelength measurements in the second spectral order." That remark could be deleted. Just explain what you used, not what you didn't use.

line 143 "The maximum signal-to-noise (SNR) ratio for a single pixel spectrophotometric light curve is 617..." You should probably clarify that the SNR is for the total signal (star+planet). Non-specialist readers might otherwise assume the SNR is for the planet only.

line 185, for comparison it would also be helpful to quote the H₂O log mixing ratio expected for exactly solar metallicity.

line 215, I suggest quoting the Sun's C/O value (and source) for reference.

Lines 310-312. The reader needs additional motivation for Equations (1) and (2), i.e. explain in words what you're doing here.

Lines 312 and 314. Mathematicians will be fascinated by your claim that you're setting numerical values to equal infinity. Infinity isn't a number. Do you mean that you set the error level to a very large finite value?

Line 338, can you comment on why a 4th order polynomial is adopted?

Line 473 discusses retrievals, but atmospheric retrievals are the topic of the section beginning on line 612. I think the line 473 discussion is only for the purpose of deriving the eclipse depths versus wavelength, not atmospheric properties. Please make that clearer, and maybe don't use the word "retrievals" when you're not extracting atmospheric properties.

Extended Data Figure 5, the b panel - it looks like the data at the longest bins fall below the theoretical minimum (square-root) line. Probably that's still within the error envelope. These rms values have their own error bars, so I suggest plotting an error envelope (e.g., red dotted lines) to make it clear that you're not overfitting the data, or there isn't autocorrelation. Moreover, the caption refers to hours, and also to 75 minutes, whereas the plot axis is in seconds. I suggest being kind to the reader and using either hours, minutes, or seconds, but not all three.

line 559. Please quote the values of the Gelman-Rubin statistic that you achieve.

Concerning the atmospheric retrievals, I am puzzled by the "2" that appears on the RHS of equation 9. Emergent flux is π times the integral of intensity times μ , there's no "2" (you're not merely multiplying by $2 \cdot \pi$ steradians). The form you have would be correct if μ in the sum varies from 0 to 1 and I is symmetric (not $\mu = -1$ to $+1$ as per convention). I suggest clarifying the range of the summation on the RHS.

The retrievals don't include clouds, presumably because this planet is too hot. But you should probably say that you neglect clouds for that reason.

Table 2 needs improvement. Non-specialist readers won't know about the limb darkening coefficients, so please mention those in the caption. Also the transit time should be labeled as BJD(TDB) if indeed it is.

Referee #2 (Remarks to the Author):

The paper presents the dayside thermal emission spectrum of the ultra-hot Jupiter WASP-18b using JWST's NIRISS/SOSS instrument covering 0.85 to 2.85 microns with an average resolution of 400 as part of the JWST Early Release Science Program. Three strong water features are detected with high significance and a possible detection of H₂, TiO, and VO is described, and a thermal inversion is required to fit the data (as alluded to in previous works). The metallicity of the atmosphere is found to be close to that of the Sun (consistent with previous works), which makes WASP-18b consistent with the trend of decreasing metallicity with planet mass, as observed in the Solar System, given that WASP-18b has a mass of 10 times that of Jupiter. A C/O ratio of less than unity is found (consistent with previous works) and a dayside temperature map is presented with the hot spot situated close to the sub-stellar point, consistent with previous findings based on HST data.

The paper is well written, and the methodologies are presented clearly and in detail (in the Methods section). The observations represent an important first step in detailing the atmospheres of exoplanets with JWST in the coming years and demonstrating the capabilities of this impressive observatory.

When reading the manuscript, it is not exactly clear which of the findings are new and the direct result of the JWST data and which are confirmations of results from previous works. It would be a good idea to clearly reference previous results and make more clear which of the findings in this paper are novel and not confirmations of previous findings (or confirmations of more tentative previous findings), including in the abstract. Just as an example, Gandhi et al. 2020 mention "signs of a thermal inversion" and Brogi et al. 2022 find various metallicity estimates close to solar with various assumptions.

The authors reduce the data with four independent JWST pipelines and find results consistent at

the one sigma level. It is commendable that the authors use four different pipeline to analyze the data, particularly in this early stage of JWST where the importance of data reduction approaches needs to be explored. However, I am missing a bit more interpretation on the different results from these four pipelines, apart from the one-sentence statement that they agree on average at the one sigma level. For example, in Extended Data Fig. 4, top panel, it appears that "supreme-SPOON" pipeline is systematically lower than the "nameless" pipeline redwards of 2 microns. Also, the nirHiss reduction looks significantly more noisy than the others particularly blueward of 1.4 microns – why is that? Probably, these effects constitute a minor issue and may not influence the interpretations, but a bit more quantitative interpretation on the difference between the pipelines seems warranted in these early stages of JWST and since that the team went through the efforts of reducing the data with four different pipelines.

While it is great to see the impressive results from JWST, this paper, along with some of the other ERS papers showing the first results from JWST, constitute important steps forward in the understanding of exoplanetary atmospheres, but the scientific impact of the results seem somewhat underwhelming for a Nature paper. It seems like the series of ERS papers are submitted to Nature because they represent the first results from JWST and not so much because they constitute significant scientific breakthroughs. I highly commend the efforts of the ERS team, which are very valuable for the community, and the efforts are certainly important and worthwhile, in particular in assessing the suitability of the different JWST instruments for atmospheric observations, but it seems to me that such papers would be better suited for an astrophysical journal rather than Nature. The many details on data reduction, treatment, and analysis have to go in the Methods section of the paper, which is a bit unfortunate.

The author list comprises a massive 76 co-authors. It is therefore likely that a fraction of the co-authors had little direct involvement in analyzing and interpreting the data and writing the paper, other than perhaps attending a number of telecons or participating in the initial design of the program. This suspicion seems to be strengthened by a quick tallying of the number of authors mentioned in the author contribution section, which reveals that only about 39 of the 76 co-authors are mentioned explicitly. Furthermore, a number of these 39 co-authors are only listed as having contributed to things like the "design of the program" and "pre-launch data challenges" and thus do not seem to have directly contributed to the analysis, interpretation, and writing of the publication. These comments may be particularly relevant to some of the more senior co-authors, who, due perhaps more to their status than their actual direct contribution, may be enjoying co-authorship on many of these types of papers and thus may be "diluting" the authorship role of early-career scientists who may have made more direct contributions to the paper. The ERS team could have chosen to only add co-authors who actually made direct and significant contributions to the papers. I want to emphasize that these comments are directed at the more senior members of the ERS team and not at the early career first author of the paper.

The authors state that all the retrieval methods use the same PHOENIX stellar model ($T_{\text{eff}} = 6435$ K, $\log g = 4.35$, and $[\text{Fe}/\text{H}] = 0.1$) to convert the model planet flux spectra to F_p/F_s values. The authors choose to use a model spectrum rather than the observed stellar spectrum "to avoid the possible introduction of systematic errors in the through the process of absolute flux calibration." (Side note: check typo in this sentence). Deriving precise stellar parameters and modeling stellar spectra has its own potential systematic issues, which could also introduce systematics in the interpretation. The authors provide only this single sentence regarding issues with the absolute flux calibration and since this precludes using the observed stellar spectrum (and thus avoiding the potential issues with stellar parameters and models), a more detailed description of the issues with absolute flux calibration seems warranted.

Likewise, since all the retrievals use the same stellar model, it would be interesting to explore if using different stellar models (e.g. ATLAS or MARCS models) would have any implications on the interpretation of the retrievals. Likewise, it would be interesting to see a discussion of what impact different stellar parameters (variations within their formal uncertainties and effects of more systematic errors) would have on the results (if any). A comparison where the team tried to

employ an absolute flux calibration to utilize the observed stellar spectrum with the results from using the PHOENIX model would also be welcomed.

Finally, the authors mention a potential tension between their results and earlier works, but advocate alleviating this tension by considering that TiO and water thermally dissociate in the upper atmosphere:

"Our best fit radiative-convective model provides strong evidence that the temperature inversion is caused by the absorption of stellar light by TiO (see Extended Data Fig. 7). At first sight this can seem at odds with high-spectral resolution observations that have detected other species able to create thermal inversions, such as atomic iron 10, but have had trouble detecting TiO 25. This tension is easily solved when considering that both TiO and water thermally dissociate in the upper atmospheric layers of ultra-hot Jupiters."

This seems like a plausible explanation, but it would be quite interesting to see a combined analysis of existing data on WASP-18b with the JWST data. This is likely beyond the scope of this paper, so this is just meant as a comment.

Response to comments by the referee:

In what follows, we respond point-by-point to each comment by the referees. *The original comments by the referees are in blue italic font.* Our response is in *black font*. Where appropriate, we included screenshots from the manuscript to facilitate the review process. We list additional refinements to the manuscript at the bottom of this document.

Referee #1 (Remarks to the Author):

This paper describes important and well executed results that will be of wide interest. The data and the analysis are of generally high quality, and I have no show-stopping criticisms. But this paper has an impressive amount of substance, and I think serious consideration should be given to dividing it into two papers - one paper dealing with the emergent day side spectrum and composition, and a second paper describing the map and implications for atmospheric drag. Both of those papers would (in my opinion) be sufficiently important to warrant publication in Nature. And two papers would be much more focused, readable, and citable, than one paper.

We thank the referee for this suggestion. We think that further investigation of the retrieved map and its implications for atmospheric drag requires the application of spectroscopic eclipse mapping, which is work that is currently in progress. We have also consulted with the editor and they have suggested a “long” format article, which provides us with the space needed to discuss our results. We therefore think it is best to keep the current work as a single paper and leave a more in depth interpretation of the dynamics for future work.

Given the ample substance in this paper, I have quite a few issues that the authors should consider before the final version(s) is/are accepted:

Lines 139, 410, and 468. I question whether fitting for the shape and duration of the eclipse is justified, as opposed to freezing those parameters based on TESS observations of the transit. The transit being a much stronger signal, and the TESS data being of high quality over many transits, it is arguably preferable to simply freeze the orbital parameters at their TESS values instead of fitting for them using the eclipse (even with narrow priors). On the other hand, the JWST data have exquisite precision, so maybe the authors are OK in this case, but that’s not obvious. Moreover, the orbital parameters in Extended Table 2, and those quoted in line 468 aren’t identical, albeit they are consistent within the errors. But it’s the same planet in the same orbit, so why not constrain the exact same orbital parameters in all cases? I suggest that the authors should reconsider the fitting methods, and arguably the best and most consistent procedure would be to fit the TESS transit data and the JWST data constrained with the exact same orbital parameters - including the white light curve, the spectral eclipses, and the eclipse map. Or at a minimum, clarify the process by summarizing how all of those fitted parameters relate to the TESS data that seem to be the primary source of the orbital parameters.

Thank you for this comment. We agree that we can describe our choice of treatment of the orbital parameters in more detail. The referee is correct that the decision to use the orbital parameters derived from the TESS data was made due to the much higher sensitivity of transit observations to the semi-major axis and impact parameter. Those observations also have the advantage that there is no potential correlation between a non-uniform atmospheric distribution and the derived orbital parameters, unlike with secondary eclipse observations. We however chose to leave those two parameters free in the white light curve fit (with gaussian priors centered on the TESS values and using the retrieved 1 sigma uncertainties as the standard deviation) to ensure that the values retrieved from the white light secondary eclipse fit were marginalized over the values of semi-major axis and impact parameter within the TESS uncertainties. This results in values of a/R_s and b retrieved from the NIRISS/SOSS white light curve that are within the 1 sigma uncertainties of the TESS analysis. We note that such a deviation from the median retrieved TESS values do not affect the retrieved spectra and eclipse map. We have discussed the points above in more detail in the paper, with the updated sections starting at lines 441 and 505. We have also updated the Extended Data Table 2 to include the fixed and fitted parameters in the NIRISS/SOSS white light curve, which clarifies the process we went through for the secondary eclipse fit.

A major component that is missing from this paper is the timing, duration and depth of the secondary eclipse as observed by JWST. There's a Table for TESS results, but oddly there's no Table describing the eclipse parameters from JWST. Please list the JWST secondary eclipse time and duration in a prominent Table (and including the white-light depth and other parameters as appropriate).

This is a good point. We have addressed this by including a second column in Extended Data Table 2 which summarizes the retrieved and fixed parameters from the NIRISS/SOSS white light curve fit. The updated table is shown below.

Parameter	TESS	JWST NIRISS/SOSS
R_p/R_*	0.09783 ± 0.00028	0.09783
T_0 [BJD _{TDB}]	$2458747.985032 \pm 0.000027$	–
T_{sec} [BJD _{TDB}]	–	$2459802.381867 \pm 0.000092$
P [days]	$0.941452382 \pm 0.000000069$	0.941452382
b	0.360 ± 0.026	0.340 ± 0.018
a/R_*	3.496 ± 0.029	3.483 ± 0.021
u_1	0.290 ± 0.032	–
u_2	0.169 ± 0.061	–
\bar{f}_p [ppm]	180 ± 13	–
F_{atm} [ppm]	177.5 ± 5.7	–
D [ppm]	21.8 ± 5.2	21.8
E [ppm]	172.2 ± 5.6	172.2
Eclipse depth [ppm]	357 ± 14	1096 ± 38
Nightside flux [ppm]	2 ± 14	–
i [deg]	84.09 ± 0.47	84.39 ± 0.30

Extended Data Table 2 | TESS and JWST NIRISS/SOSS fit results. The median and 1σ uncertainties of the astrophysical parameters from the TESS (left column) and JWST NIRISS/SOSS (right column) analyses of WASP-18. The transit observations in the TESS phase curve are fitted considering the u_1 and u_2 quadratic limb-darkening coefficients. In our TESS phase curve parameterization, the secondary eclipse depth and nightside flux are derived parameters calculated from the average relative planetary flux \bar{f}_p and the planet’s phase curve semi-amplitude F_{atm} . Likewise, the orbital inclination i is derived from the scaled semi-major axis a/R_* and the impact parameter b . The parameters fixed to the values from the TESS analysis for the NIRISS/SOSS light curve fitting are shown without uncertainties in the table. The retrieved parameters are the time of secondary eclipse T_{sec} , the impact parameter b , and the semi-major axis a/R_* , the latter two being fitted considering normal priors from the TESS constraints. The NIRISS/SOSS eclipse depth is derived from the samples of the astrophysical model (Eq. 4).

When discussing the metallicity of the planet (your results are highly interesting), it would be appropriate to consider what you could say about possible limits on accretion during migration, quoting models as might be appropriate. This planet has evidently migrated a long distance, and it seems surprising that it didn’t eat many comets and asteroids along the way. Can the authors usefully comment on that issue.

This is a good suggestion. We now discuss the constraint on the mass of accreted metals from our atmospheric metallicity measurement in the main text and methods, as shown below.

228 finding of solar metallicity, three times lower than that of Jupiter, is consistent with this trend,
229 given WASP-18b’s mass of $10.4 M_J$. Assuming it formed at solar metallicity, we find that up to
230 $181 M_{\oplus}$ of metals could have been accreted during WASP-18b’s migration before it exceeds the 2σ
231 upper limit on the metallicity obtained from the atmospheric retrieval (see Methods). This quantity
232 of metals almost certainly exceeds the amount that is available in the disk for accretion during
233 migration. The metallicity that is measured for WASP-18b is therefore likely to be representative
234 of the bulk composition of the protoplanetary disk at the planet’s formation location. The low C/O

725 Due to WASP-18b’s large mass of $M_p = 10.4 M_J$, an important amount of rocky and icy
726 material can be accreted without significantly changing the overall metallicity of the planet. As
727 a zeroth order estimate, we assume that the planet formed with exactly solar metallicity. Then,
728 the mass of metals accreted needed to increase the overall metallicity to N times solar is given
729 by $Z_{\odot} M_p (N - 1)$, where $Z_{\odot} = 0.0134^{27}$ is the solar metal mass fraction. From this relation and
730 assuming that the planet’s envelope is well-mixed, we relate our retrieved metallicity probability
731 posterior to the mass of metals accreted.

Line 228, in the brightness-temperature mapping, the authors assume that the star emits as a blackbody. That’s a poor assumption, and that portion of the analysis should be re-done using model atmospheres. Interpolating in a model atmosphere grid for the star is a much better approximation than a blackbody star. There’s no excuse for using a blackbody when high quality model atmospheres are available.

The referee is correct that it is more accurate to use stellar model atmospheres rather than a simple blackbody to convert the Fp/Fs maps to brightness temperature maps. This portion of the analysis has been re-done using PHOENIX atmosphere models and the results starting on line 254, as well as figure 4, have been updated accordingly. The methodology used to perform this conversion has been updated in the paper, as shown below.

584 The resulting weights of each eigencurve are then applied to the corresponding eigenmaps to
585 generate a flux map of the planet. We convert the star-normalized flux map to brightness tempera-
586 ture by assuming the planet is a blackbody and the star emits as a PHOENIX⁴¹ spectrum calculated
587 with PyMSG⁷⁹, both integrated over the NIRISS/SOSS throughput. We estimate temperature map
588 uncertainties by computing a subsample of maps from the MCMC posterior distribution and cal-
589 culating 68.3%, 95.5%, and 99.7% quantiles at each location, including the effects of uncertainties
590 in planetary radius, stellar temperature and stellar $\log(g)$.

In the paragraph beginning on line 596, the discussion of the orbital eccentricity is confusing. The timing of the eclipse is more sensitive to the eccentricity than the RV observations are. Already there were strong limits placed on the eccentricity ($e \cdot \cos\omega$) by Nymeyer et al. and Maxted et al. and indeed the timing and duration of this JWST eclipse combined with photometry from TESS should provide very stringent limits on the eccentricity. Instead of quoting a possible eccentricity from RV measurements, I don’t understand why you don’t derive and use strong limits on the eccentricity from the TESS, Spitzer, and your own JWST photometry. Indeed, line 414 already said that you have a strong preference for zero eccentricity.

That is a good point. Strong constraints were already placed on the eccentricity from earlier studies using photometric observations (notably Triaud 2010 and Nymeyer 2011 both finding a retrieved eccentricity of ~ 0.009). These studies are now referred to when discussing the possibility of a non-zero eccentricity. The referee is correct that our justification for using a circular orbit comes from the preference for the 0 eccentricity case in the TESS analysis. Unfortunately, the amount of time elapsed between the last observed TESS transit and our NIRISS/SOSS secondary eclipse (~ 1000 days) largely inflates the uncertainty on the TESS transit time when comparing it to our retrieved eclipse time. This leads to a relatively poor constraint of $e \cdot \cos\omega$ compared to past studies. Therefore, to address the non-zero eccentricity measurement from past studies, we show that such deviations from a circular do not affect our retrieved map. The updated text is shown below.

644 statistical preference. However, all resulting maps were well within the uncertainties of one of the
645 two models presented. We note that, while the eccentricity is kept fixed to zero throughout this
646 analysis, as justified by the preference for a circular orbit in the TESS analysis, considering an
647 eccentric orbit would allow to first order for variations in mid-eclipse time and eclipse duration⁸³.
648 Past photometric and radial velocity observations of WASP-18b have found a small but non-zero
649 eccentricity for WASP-18b on the order of $e = 0.008$ ^{84;85;86}, corresponding to an offset of the time
650 of mid-eclipse of 9 seconds, as well as a difference of 120 seconds between the transit and eclipse
651 durations. These differences in eclipse timing and duration are of the same magnitude as those

Line 542, “it is possible to recover a fit which contains regions of negative planet emission. This is physically impossible, so we impose a positivity constraint on the total flux map.” I am concerned that a positivity constraint will bias the eclipse maps, due to Lucy-Sweeny bias. Can the authors justify quantitatively that the results will not be biased?

This is a good point to address. We have added a discussion of the Lucy-Sweeney bias in the eclipse mapping methods and show that the results are not sensitive to the flux lower limit allowed when fitting the map as no position on the planet approaches negative flux values. The updated section is shown below.

566 total flux map. While the eigenmaps are mathematically defined across the entire planetary sphere,
567 our observations only constrain the portion of the planet that is visible during the observation, so
568 we only enforce the flux positivity condition in the visible region of the planet. While this positivity
569 condition could introduce a Lucy-Sweeney⁷⁷ bias near zero flux, we note that our fitted maps (and
570 GCM predictions) are far from negative and increasing this boundary to, for example, 300 K leads
571 to no change in results. We test all combinations of $l_{\max} \leq 6$ and $N \leq 8$ using a least-squares

The eclipse map (Figure 4) is done in white-light, but that bandpass has an effective wavelength. I suggest quoting the effective wavelength. I suggest quoting the effective wavelength because JWST maps will doubtless be made in other bands. (the effective wavelength is derived from integrating the emergent flux weighted by the instrument sensitivity.)

Thank you for this suggestion, the effective wavelength is indeed important to understand which wavelength domain of the thermal emission contributes most to the observed map. We have computed the effective wavelength by integrating the wavelengths observed by NIRISS/SOSS

weighted by the system flux (star + planet) and instrument response and find it to be $\lambda_{eff} = 1.27 \mu\text{m}$, which is now quoted in the caption of Figure 4.

Minor comments:

First paragraph, “whereas the SUBSTRIP256 subarray also provides the shorter-wavelength in the second spectral order”. That remark could be deleted. Just explain what you used, not what you didn’t use.

That is a good point. This remark has been taken out of the sentence which now reads: “The SUBSTRIP96 mode covers the first spectral order between 0.85 and 2.85 μm ”.

Line 143, “The maximum signal-to-noise (SNR) ratio for a single pixel spectrophotometric light curve is 617...”. You should probably clarify that the SNR is for the total signal (star+planet). Non-specialist readers might otherwise assume the SNR is for the planet only.

Good idea. We have updated this sentence to: “The maximum star plus planet signal-to-noise (SNR) ratio for a single pixel..”.

Line 185, for comparison it would also be helpful to quote the H2O log mixing ratio expected for exactly solar metallicity.

We agree this is a helpful comparison. We now quote the deep interior solar H2O log mixing ratio value of -3.21 ($\log_{10}[6.1 \times 10^{-4}]$) for comparison with the retrieved value of -3.23.

Line 215, I suggest quoting the Sun’s C/O value (and source) for reference.

Good point, we have updated the sentence on line 195, where the C/O constraints are first discussed, so that the solar C/O value (0.55) is quoted with the relevant source.

Lines 310-312, the reader needs additional motivation for Equations (1) and (2), i.e. explain in words what you’re doing here.

Thank you for bringing up the need for further detail. We now describe in words the motivation behind eq. 1 and 2, as shown below.

308 that manifests as a sudden decrease in flux. First, we compute the median columns \bar{c} before and
 309 after the tilt event; we use integrations 300–900 and 1350–1900. Then, we define a given column
 310 j and row k at an integration i as the sum of the scaled median column $m_{i,j}\bar{c}_{j,k}$ and the $1/f$ noise
 311 $n_{i,j} = c_{i,j,k} - m_{i,j}\bar{c}_{j,k}$. Using the errors $\epsilon_{i,j,k}$ returned by the jwst pipeline, we solve for the
 312 values of $m_{i,j}$ and $n_{i,j}$ that minimize the chi-square between the observed and the scaled columns
 313 $\chi^2 = \sum_k [(c_{i,j,k} - m_{i,j}\bar{c}_{j,k} - n_{i,j})/\epsilon_{i,j,k}]^2$. These values of the scaling $m_{i,j}$ and $1/f$ noise $n_{i,j}$ are
 314 obtained by imposing $\partial\chi^2/\partial m_{i,j}, \partial\chi^2/\partial n_{i,j} = 0$, such that

Lines 312 and 314, mathematicians will be fascinated by your claim that you're setting numerical values to equal infinity. Infinity isn't a number. Do you mean that you set the error level to a very large finite value?

That is correct. We have updated the sentence starting on line 338 to clarify that what is considered infinity in this case is the IEEE 754 floating point representation of positive infinity, which is the definition given by numpy for the numpy.inf value.

Line 338, can you comment on why a 4th order polynomial is adopted?

We now explain our choice of a 4th order polynomial to fit the trace in the sentence starting on line 365. The choice was made by visually inspecting the quality of the fit to the trace and we find that a 2nd order polynomial tends to underfit while a 6th order function usually overfits and introduces wiggles in the trace profile. A 4th order function proves flexible enough while keeping the shape of the trace smooth.

Line 473 discusses retrievals, but atmospheric retrievals are the topic of the section beginning on line 612. I think the line 473 discussion is only for the purpose of deriving the eclipse depths versus wavelength, not atmospheric properties. Please make that clearer, and maybe don't use the work "retrievals" when you're not extracting atmospheric properties.

Thank you for noticing this. We have made sure that the term retrieval is only employed when discussing atmospheric properties in the paper. The sentence on line 473 now reads: "Light curve fits are performed using the Affine Invariant ...".

Extended Data Figure 5, the b panel - it looks like the data at the longest bins fall below the theoretical minimum (square-root) line. Probably that's still within the error envelope. These rms values have their own error bars, so I suggest plotting an error envelope (e.g., red dotted lines) to make it clear that you're not overfitting the data, or there isn't autocorrelation. Moreover, the caption refers to hours, and also to 75 minutes, whereas the plot axis is in seconds. I suggest being kind to the reader and using either hours, minutes, or seconds, but not all three.

We thank the referee for this suggestion. We have updated Extended Data Figure 5 to include 1 sigma error envelopes around the theoretical Poisson noise. We have also modified the axis and

caption so that all values are in units of hours. For the spectroscopic light curves, it does seem that there is a decrease in the normalized RMS outside of the expected variance. We note that similar trends have been observed in the NIRC*am* and NIRSpec/G395H analyses of the WASP-39b transit observations (shown in the two figures below, the first being from Ahrer et al., 2023 and the second from Alderson et al., 2023). We note that Allen variance plots were not shown in the NIRSpec/PRISM and NIRISS/SOSS WASP-39b papers. Because this trend is observed in all works published thus far and since the complexity of the models considered for the light curve fitting of the WASP-39b transit observations was somewhat minimal, we think this decrease might be caused by autocorrelation in the data that is common to the different instruments and modes of JWST. Our updated figure and caption are shown after the figures from Ahrer+2023 and Alderson+2023.

Extended Data Figure 4: Normalised root mean square error as a function of bin size for all spectroscopic channels. The red line shows the expected relationship for perfect Gaussian white noise. The black lines show the observed noise from each spectroscopic channel for the Eureka! long wavelength reduction. Values for all channels are normalized by dividing by the value for a bin size of 1 in order to compare bins with different noise levels. The black lines closely follow the red line out to large bin sizes of ≈ 30 (≈ 0.5 -hr time

Extended Data Figure 4: Normalised root-mean-squared binning statistic for three of the 11 reductions detailed in Methods. In each subplot, the red line shows the expected relationship for perfect Gaussian white noise. The black lines show the observed noise from each spectroscopic light curve for pipelines 1, 3, and 5. In order to compare bins and noise levels, values for all bins in each pipeline are normalised by dividing by the value for a bin width of 1.

Extended Data Fig. 5 | Light curve residuals binned in time. **a**, Absolute root mean square (RMS) of the residuals as a function of bin size (black line) for the white light curve. The RMS values are plotted against the Poisson noise limit (red line), which goes down as the square root of the number of integrations contained in a single bin. We also show the theoretical 1σ error envelope of the Poisson noise. The residuals bin down to ~ 5 ppm for bins of 1 hour. The broadband residuals do not follow perfectly the Poisson noise, which is indicative of remaining time correlations. **b**, Normalized RMS of the 408 spectrophotometric light curves considered in the analysis. We observe that the residuals follow the Poisson noise limit from bin sizes of a single integration up to bins of ~ 1 hour, indicating that there are no time correlations in the residuals. We observe a slight decrease of the normalized RMS below the Poisson noise at larger bin sizes, similar to what was observed in the NIRCcam and NIRSpc/G395H observations of WASP-39b^{22;162}.

Line 559, please quote the values of the Gelman-Rubin statistic you achieve.

The sentence now states that the Gelman-Rubin statistics achieved are all equal to or below 1.00006, which satisfies the convergence criterion.

*Concerning the atmospheric retrievals, I am puzzled by the “2” that appears on the RHS of equation 9. Emergent flux is pi times the integral of intensity times mu, there’s no “2” (you’re not multiplying by 2*pi steradians). The form you have would be correct if mu in the sum varies from 0 to 1 and I is symmetric (not mu = -1 to +1 as per convention). I suggest clarifying the range of the summation on the RHS.*

That is a good point. The interval of the sum in equation in equation 9 has been clarified by starting from the integral over mu, which is considered to range here from 0 to 1. The range of summation of mu ($\mu \in [0,1]$) is specified once more in the sentence following eq. 9.

The retrievals don’t include clouds, presumably because this planet is too hot. But you should probably say that you neglect clouds for that reason.

The referee is correct that the retrievals presented in this work do not include clouds as they are not expected to condense at WASP-18b’s dayside temperatures. This is now stated in the introduction of the Atmospheric Retrieval methods section (line 661) referencing Gao et al. 2020.

Table 2 needs improvement. Non-specialist readers won’t know about the limb-darkening coefficients, so please mention those in the caption. Also the transit time should be labeled as BJD(TDB) if indeed it is.

Good point. As discussed in one of the responses above, we have updated Extended Data Table 2 and now mention the limb-darkening coefficients as well as the transit time unit.

Referee #2 (Remarks to the Author):

When reading the manuscript, it is not exactly clear which of the findings are new and the direct result of the JWST data and which are confirmations of results from previous works. It would be a good idea to clearly reference previous results and make more clear which of the findings in this paper are novel and not confirmations of previous findings (or confirmations of more tentative previous findings), including in the abstract. Just as an example, Gandhi et al. 2020 mention “signs of a thermal inversion” and Brogi et al. 2022 find various metallicity estimates close to solar with various assumptions.

We thank the referee for this suggestion. To address this, we have added a paragraph at the beginning of the paper that situates our work in the wider context of ultra-hot Jupiters and also gives an overview of the results that have been obtained from prior observations of WASP-18b. This new paragraph is shown below.

106 The thermal emission spectra of ultra-hot Jupiters typically have muted spectral features and
107 thus closely resemble blackbodies in existing narrowband measurements^{2;3}. The interpretation of
108 these spectra has been controversial, with some studies claiming that the data are indicative of high
109 metallicities and C/O ratios^{4;6}. Alternatively, other studies have proposed that approximately solar-
110 composition models including the effects of molecular dissociation and continuum opacity from
111 the H⁻ ion can match the data^{7;8;9;10;14}. The ultra-hot Jupiter WASP-18b has been a subject of this
112 controversy. Past Hubble and Spitzer Space Telescope secondary eclipse and phase curve obser-
113 vations have found high dayside temperatures^{15;16}, indicative of low heat redistribution potentially
114 caused by magnetic drag¹⁶, weak or no spectral features from H₂O in the Hubble bandpass, and
115 signs of a temperature inversion in the broadband Spitzer photometry^{4;7;17}. High-resolution obser-
116 vations of WASP-18b’s dayside have detected CO, OH, and H₂O at signal-to-noise ratios of 4.0,
117 4.8, and 3.3 respectively¹⁸. The molecular features were observed in emission, indicative of a ther-
118 mal inversion, although the lack of a spectral continuum led to poor constraints on the temperature
119 of the atmosphere. The metallicity and C/O ratio values retrieved from the high-resolution data are
120 consistent with solar but also depend on the physical assumptions, with the metallicity constraints
121 ranging from 1–100 times solar between the self-consistent and free chemistry analyses.

The authors reduce the data with four independent JWST pipelines and find results consistent at the one sigma level. It is commendable that the authors use four different pipelines to analyze the data, particularly in this early stage of JWST where the importance of data reduction approaches needs to be explored. However, I am missing a bit more interpretation on the different results from these four pipelines, apart from the one-sentence statement that they agree on average at the one sigma level. For example, in Extended Data Fig. 4, top panel, it appears that “supreme-SPOON” pipeline is systematically lower than the “nameless” pipeline redwards of 2 microns. Also the nirHiss reduction looks significantly more noisy than the others particularly blueward of 1.4 microns - why is that? Probably, these effects constitute a minor issue and may not influence the interpretations, but a bit more quantitative interpretation on the difference between the pipelines seems warranted in these early stages of JWST and since that the team went through the efforts of reducing the data with four different pipelines.

We agree with the referee that it is valuable to go into further detail on the differences in the spectra that arise from the four reductions. We’ve addressed this comment by running the same SCARLET retrieval analysis on all four reductions to evaluate the impact of those differences on our inferred atmospheric parameters. We also state the effects that are most likely causing the differences between reductions. Below is the paragraph we have added to address this comment.

732 We also quantify the impact of the choice of reduction on the retrieved atmospheric prop-
733 erties by performing the same retrieval on the four spectra shown in Extended Data Fig. 4. We
734 find that all reductions retrieve metallicities that are within 1σ of the NAMELESS reduction, with
735 $[M/H]$ values of $0.00_{-0.66}^{+0.38}$, $0.05_{-0.33}^{+0.30}$, and $0.37_{-0.31}^{+0.38}$ for the `nirHiss`, `transitspectroscopy`,
736 and `supreme-SPOON` reductions respectively. We also retrieve C/O 3σ upper limits of 0.749,
737 0.602, and 0.627 in that same order. We note that the slightly higher metallicity retrieved from the
738 `supreme-SPOON` reduction is most likely due to the downward slope longward of $2\ \mu\text{m}$, possibly
739 caused by dilution of the signal through the process of background subtraction or $1/f$ correction,
740 which is not observed in the `nirHiss` and `transitspectroscopy` reductions. We also find
741 that the `nirHiss` retrieves larger uncertainties on the measured $[M/H]$ and C/O, caused by the
742 larger scatter at short wavelengths, which is possibly introduced through the optimal extraction
743 method as this is not seen in the reductions using box extraction.

While it is great to see the impressive results from JWST, this paper, along with some of the other ERS papers showing the first results from JWST, constitute important steps forward in the understanding of exoplanetary atmospheres, but the scientific impact of the results seem somewhat underwhelming for a Nature paper. It seems like the series of ERS papers are submitted to Nature because they represent the first results from JWST and not so much because they constitute significant breakthroughs. I highly commend the efforts of the ERS team, which are very valuable for the community, and the efforts are certainly important and worthwhile, in particular in assessing the suitability of the different JWST instruments for atmospheric observations, but it seems to me that such papers would be better suited for an astrophysical journal rather than Nature. The many details on data reduction, treatment, and analysis have to go in the Methods section of the paper, which is a bit unfortunate.

While this comment is mostly addressed to the Editor, we offer these scientific highlights that help justify a high-impact paper in this specific case:

- **A definitive resolution of the controversy surrounding our understanding of the muted spectra of ultra-hot Jupiters.** We have confidently detected multiple subtle water emission features and we show that thermal inversions together with molecular dissociation sculpt the spectrum of an archetype of this class of planet. We also determine the planet's atmospheric metallicity and place an upper limit on its C/O ratio, showing that these are in line with what is roughly expected from planet formation models and not highly unusual as some previous studies had claimed.
- **A high S/N eclipse map of the planet that indicates an unexpected brightness distribution over the dayside of the planet.** This is only the second planet with an eclipse map. This is the first eclipse map of an ultra-hot Jupiter, which is a class of planets for which the atmospheric circulation is thought to be impacted by the magnetic field. Indeed, our leading explanation for the observed brightness distribution on WASP-18b is a global magnetic field with a strength of 5 G or larger. There are almost no other observational constraints on the magnetic field strengths of extrasolar planets.

The author list comprises a massive 76 co-authors. It is therefore likely that a fraction of the co-authors had little direct involvement in analyzing and interpreting the data and writing the paper, other than perhaps attending a number of telecons or participating in the initial design of the

program. This suspicion seems to be strengthened by a quick tallying of the number of authors mentioned in the author contribution section, which reveals that only about 39 of the 76 co-authors are mentioned explicitly. Furthermore, a number of these 39 co-authors are only listed as having contributed to things like “the design of the program” and “pre-launch data challenges” and thus do not seem to have directly contributed to the analysis, interpretation, and writing of the publication. These comments may be particularly relevant to some of the more senior co-authors, who, due perhaps more to their status than their actual direct contribution, may be enjoying co-authorship on many of these types of papers and thus may be “diluting” the authorship role of early-career scientists who may have made more direct contributions to the paper. The ERS team could have chosen to only add co-authors who actually made direct and significant contributions to the papers. I want to emphasize that these comments are directed at the more senior members of the ERS team and not at the early career first author of the paper.

We acknowledge the referee’s comment but there is no action to take. As described in the author contribution section, everyone listed made sufficient contributions to be co-authors on the paper according to standard practice in astronomy. For example, designing the program and developing algorithms on simulated data were essential to the success of this project. Our collaboration has developed a publication plan with a delineated set of 13 programmatic papers that are open to everyone who made these and other similar contributions. Everyone deserves to be recognized for their work no matter what their career stage is, and these papers are the culmination of a long-term project that has had many phases. The author list is given in roughly decreasing order of contribution for this specific paper, which again, is standard in astronomy. Of the first 12 authors, 11 are grad students or postdocs. So the contributions of the more junior people are being highlighted.

The authors state that all the retrieval methods use the same PHOENIX stellar model ($T_{\text{eff}} = 6435$ K, $\log g = 4.35$, and $[\text{Fe}/\text{H}] = 0.1$) to convert the model planet flux spectra to F_p/F_s values. The authors choose to use a model spectrum rather than the observed stellar spectrum “to avoid the possible introduction of systematic errors through the process of absolute flux calibration.” (Side note: check typo in this sentence). Deriving precise stellar parameters and modeling stellar spectra has its own potential systematic issues, which could also introduce systematics in the interpretation. The authors provide only this single sentence regarding issues with the absolute flux calibration and since this precludes using the observed stellar spectrum (and thus avoiding the potential issues with stellar parameters and models), a more detailed description of the issues with absolute flux calibration seems warranted.

This is a good point, it is worthwhile to discuss the issues we found with absolute flux calibration as this also applies to other NIRISS/SOSS observations. This comment is addressed in the new paragraph shown below.

722 = $0.05_{-0.25}^{+0.26}$ and C/O 3σ upper-limit of 0.603. Finally, we test the effect of using the flux calibrated
723 spectrum on the retrieved atmospheric properties. The use of a flux calibrated spectrum, measured
724 directly from the NIRISS/SOSS observations, was avoided due to some slight issues found in the
725 currently available CRDS reference files used in the `photom` step of the `jwst` pipeline Stage 2.
726 The most recent reference file, `photom_0034`, is able to reproduce accurately the continuum from
727 the PHOENIX model considered in the main retrieval but shows significant noise in the observed
728 spectrum. We also looked at reference file `photom_0037`, which was produced from ground
729 data and does not account for the larger than expected throughput that was observed on sky¹⁶.
730 Despite this, we perform a retrieval on the flux calibrated stellar spectrum obtained by smoothing
731 the response curve of reference file `photom_0034` with a median filter of width 100. The retrieval
732 ran on the flux calibrated stellar spectrum retrieves a metallicity [M/H] of $0.11_{-0.68}^{+0.24}$ and a C/O 3σ
733 upper limit of 0.739.

Likewise, since all the retrievals use the same stellar model, it would be interesting to explore using different stellar models (e.g. ATLAS or MARCS models) would have any implications on the interpretation of the retrievals. Likewise, it would be interesting to see a discussion of what impact different stellar parameters (variations within their formal uncertainties and effects of more systematic errors) would have on the results (if any). A comparison where the team tried to employ an absolute flux calibration to utilize the observed stellar spectrum with the results from using the PHOENIX model would also be welcomed.

Thank you for this suggestion, following the same methodology as discussed in the point above, we have addressed this comment in the methods section. Shown below is the new paragraph.

710 We quantify the impact of the stellar spectrum considered for the analysis on the retrieved
711 atmospheric properties by running the same retrieval while varying the stellar spectrum. First, we
712 explore the impact of the PHOENIX stellar model parameters by varying them within their 1σ
713 uncertainties (50 K for T_{eff} , 0.05 for $\log g$, and 0.1 for [M/H]). We find these variations to have
714 minimal impact on the retrieved metallicity with the measured median varying at most by 0.04 dex
715 ($\sim 0.15\sigma$), the same is true for the C/O upper-limit with all retrieved upper-limits being within 0.06
716 of the retrieval with the standard stellar parameters, with the exception of the $T_{eff} = 6385$ K case
717 which retrieves $C/O < 0.64$ at 3σ but does not affect our conclusions on WASP-18b’s formation
718 and migration history. In second place, we test the impact of the type of stellar model considered
719 by also running the retrieval with an ATLAS9 stellar model¹¹¹ ($T_{eff} = 6435$ K, $\log g = 4.35$,
720 [M/H] = 0.1). We find that the effect on the results are similar to those observed when varying the
721 stellar parameters within their 1σ uncertainty, with a retrieved metallicity measurement of [M/H]
722 = $0.05_{-0.25}^{+0.26}$ and C/O 3σ upper-limit of 0.603. Finally, we test the effect of using the flux calibrated

Shown below is a table summarizing the results from our retrievals considering the various stellar spectra described above.

T_{eff} [K]	$\log g$	[Fe/H]	[M/H]	C/O	χ^2/N	χ^2
6435	4.35	0.1	$0.08^{+0.24}_{-0.27}$	< 0.598	1.092	446
6385	4.35	0.1	$0.06^{+0.25}_{-0.36}$	< 0.642	1.090	445
6485	4.35	0.1	$0.07^{+0.26}_{-0.40}$	< 0.601	1.094	446
6435	4.30	0.1	$0.06^{+0.26}_{-0.35}$	< 0.604	1.093	446
6435	4.40	0.1	$0.10^{+0.26}_{-0.28}$	< 0.606	1.091	445
6435	4.35	0.0	$0.07^{+0.25}_{-0.29}$	< 0.594	1.093	446
6435	4.35	0.2	$0.04^{+0.30}_{-0.37}$	< 0.605	1.092	446
6435 (ATLAS)	4.35	0.1	$0.05^{+0.26}_{-0.25}$	< 0.603	1.086	443

Table 3: **Atmospheric constraints when varying the stellar model parameters.** We find a maximum deviation in the median retrieved metallicity of 0.04 dex ($\sim 0.2\sigma$). The maximum deviation of the 3σ C/O upper limit is of 0.04. The results are therefore robust against 1σ variations in the stellar model parameters.

Finally, the authors mention a potential tension between their results and earlier works, but advocate alleviating this tension by considering that TiO and water thermally dissociate in the upper atmosphere:

“Our best fit radiative-convective model provides strong evidence that the temperature inversion is caused by the absorption of stellar light by TiO (see Extended Data Fig. 7). At first sight this can seem at odds with high-spectral resolution observations that have detected other species able to create thermal inversions, such as atomic iron 10, but have had trouble detecting TiO 25. This tension is easily solved when considering that both TiO and water thermally dissociate in the upper atmospheric layers of ultra-hot Jupiters”.

This seems like a plausible explanation, but it would be quite interesting to see a combined analysis of existing data on WASP-18b with the JWST data. This is likely beyond the scope of this paper, so this is just meant as a comment.

We thank the referee for this comment. The study of the synergies between the JWST data and past observations will be the subject of future work.

Formatting Changes

LENGTH: With the addition of the introductory paragraph, the main text is now at ~ 2700 words which is below the 3000 words limit for the “long” category.

TITLES: It is difficult to summarize the large amount of new results into a compact title. We have therefore chosen to keep the initial title but are open to suggestions.

SUMMARY PARAGRAPH: We have kept the summary paragraph as is due to the limited amount of space and because of the in-depth overview given in the introduction paragraph.

MAIN TEXT: As discussed in the response to the referees, we have added an introduction paragraph after the summary paragraph to situate our work in the wider context of ultra-hot Jupiters and also give an overview of the results that have been obtained from prior observations of WASP-18b.

METHODS: The footnotes in the methods section have been converted to numbered references.

REFERENCES: We have reduced the number of citations in the main text to 50. We have not found a way to split the main text and methods references in the latex template but this will be done when we convert the paper to MS Word once it is accepted.

MAIN TEXT STATEMENTS: The corresponding author information is no longer given in a footnote at the beginning of the manuscript but only after the acknowledgements.

FIGURE LEGENDS: The figure legends now specify that the error bars shown for the data correspond to their 1 sigma uncertainty.

DISPLAY ITEMS: We have adjusted all main text figures to be 183 mm wide PDFs with the proper formatting.

FIGURE FORMATTING: We find all figures conform with lower case words, sans-serif font, and labeling of subpanels.

EXTENDED DATA: We have reduced the amount of Extended Data items from 12 to 10 by removing what was formerly Extended Data Fig. 1 (the reduction steps) and Extended Data Table 1 (summarizing the atmospheric constraints from the various retrievals). All extended data are properly referenced in the main text and/or methods. All figures were created with widths equal to 183 mm.

SOURCE DATA: All data presented in the figures of this manuscript will be added to our JWST ERS Zenodo repository once the manuscript is accepted.

Reviewer Reports on the First Revision:

Referee #1 (Remarks to the Author):

The authors have done a great job of responding to my comments. I can only think of one more (very minor) comment that they might want to consider before going to press. In Extended Figure 5, "RMS" is mentioned. Perhaps you mean "standard deviation"? (they're not the same).

I recommend publication in Nature; this is an excellent paper.

Referee #2 (Remarks to the Author):

The authors have taken into consideration the points I raised and carried out exhaustive tests to understand whether some of the points had any implications on the conclusions of the paper. I am therefore quite happy with their response regarding all the scientific aspects.

The only remaining point is related to the massive number of co-authors on the paper. It is unfortunate, but not surprising, that the authors have chosen not to take any action here. Obviously, anyone who makes significant and direct contributions to a paper should be recognized with co-authorship – period! If you re-read my comments, that should also be clear. However, as mentioned, only 50% of the authors (38 authors of the total of 76 authors) on the paper are explicitly mentioned in the author contribution section. Of the 38 authors mentioned in the author contribution section, 12 of these are solely listed as having contributed to definition of the proposal or other pre-launch activities, which means that 50 of the 76 authors only contributed to the design, planning, and pre-launch aspects of the program. The authors mention that a series of 13 papers is planned.

I do not intend to diminish the work that went into designing and planning the program, which undoubtedly has been a large effort, but one might question whether this work justifies co-authorship for 50 authors on 13 papers! For reference, here are the Vancouver recommendations: <https://www.icmje.org/recommendations/browse/roles-and-responsibilities/defining-the-role-of-authors-and-contributors.html>

And Nature's own guidelines state that:

"Each author is expected to have made substantial contributions to the conception or design of the work; or the acquisition, analysis, or interpretation of data; or the creation of new software used in the work; or have drafted the work or substantively revised it"

At the very least, I would suggest that all the authors on the paper are mentioned explicitly in the author contribution section, so it is possible to see what their contributions have been in the design, planning, and pre-launch phase of the program.

Apart from this, I am happy to recommend the paper for publication and I congratulate the team on a very nice paper!

Author Rebuttals to First Revision:

Response to comments by the referee:

In what follows, we respond point-by-point to each comment by the referees. *The original comments by the referees are in blue italic font.* Our response is in *black font.* Where appropriate, we included screenshots from the manuscript to facilitate the review process. We list additional refinements to the manuscript at the bottom of this document.

Referee #1 (Remarks to the Author):

The authors have done a great job of responding to my comments. I can only think of one more (very minor) comment that they might want to consider before going to press. In Extended Figure 5, “RMS” is mentioned. Perhaps you mean “standard deviation”?(they’re not the same).

I recommend publication in Nature; this is an excellent paper.

We thank the referee for their feedback. We indeed are showing the root mean square (RMS) in Extended Data Fig. 4 (formerly Extended Data Fig. 5) and not the standard deviation. The RMS metric is commonly used in Allan variance figures (e.g. Alderson+2023, Ahrer+2023).

Referee #2 (Remarks to the Author):

The authors have taken into consideration the points I raised and carried out exhaustive tests to understand whether some of the points had any implications on the conclusions of the paper. I am therefore quite happy with their response regarding all the scientific aspects.

The only remaining point is related to the massive number of co-authors on the paper. It is unfortunate, but not surprising, that the authors have chosen not to take any action here. Obviously, anyone who makes significant and direct contributions to a paper should be recognized with co-authorship - period! If you re-read my comments, that should also be clear. However, as mentioned, only 50% of the authors (38 authors of the total of 76 authors) on the paper are explicitly mentioned in the author contribution section. Of the 38 authors mentioned in the author contribution section, 12 of these are solely listed as having contributed to definition of the proposal or other pre-launch activities, which means that 50 of the 76 authors only contributed to the design, planning, and pre-launch aspects of the program. The authors mention that a series of 13 papers is planned.

I do not intend to diminish the work that went into designing and planning the program, which undoubtedly has been a large effort, but one might question whether this work justifies co-authorship for 50 authors on 13 papers! For reference, here are the Vancouver recommendations:

<https://www.icmje.org/recommendations/browse/roles-and-responsibilities/defining-the-role-of-authors-and-contributors.html>

And Nature's own guidelines state that:

“Each author is expected to have made substantial contributions to the conception or design of the work; or the acquisition, analysis, or interpretation of data; or the creation of new software used in the work; or have drafted the work or substantively revised it”

At the very least, I would suggest that all the authors on the paper are mentioned explicitly in the author contribution section, so it is possible to see what their contributions have been in the design, planning, and pre-launch phase of the program.

Apart from this, I am happy to recommend the paper for publication and I congratulate the team on a very nice paper!

We thank the referee for their feedback. To address this comment, we have made explicit the contribution/s of all authors in the author contribution section. We now list the people that have contributed to the GCM comparison, the simulation of disequilibrium chemistry models, and the work that has been done on absolute flux calibration of the stellar spectrum. Finally, we list the remaining authors, who have all provided scientific and/or technical input to the manuscript.